# Representation Equivalent Neural Operators: a Framework for Alias-free Operator Learning

**Francesca Bartolucci**[1]  **Emmanuel de Bézenac**[2]  **Bogdan Raonić**[2,3]

**Roberto Molinaro**[2]  **Siddhartha Mishra**[2,3]  **Rima Alaifari**[2,3]

1 Delft University of Technology, Netherlands
2 Seminar for Applied Mathematics, ETH, Zurich, Switzerland
3 ETH AI Center, Zurich, Switzerland

## Abstract

Recently, *operator learning*, or learning mappings between infinite-dimensional function spaces, has garnered significant attention, notably in relation to learning partial differential equations from data. Conceptually clear when outlined on paper, neural operators necessitate discretization in the transition to computer implementations. This step can compromise their integrity, often causing them to deviate from the underlying operators. This research offers a fresh take on neural operators with a framework *Representation equivalent Neural Operators (ReNO)* designed to address these issues. At its core is the concept of operator aliasing, which measures inconsistency between neural operators and their discrete representations. We explore this for widely-used operator learning techniques. Our findings detail how aliasing introduces errors when handling different discretizations and grids and loss of crucial continuous structures. More generally, this framework not only sheds light on existing challenges but, given its constructive and broad nature, also potentially offers tools for developing new neural operators.

## 1 Introduction

Operators are mappings between infinite-dimensional function spaces. Prominent examples are *solution operators* for ordinary and partial differential equations (PDEs) [9] which map function space inputs such as initial and boundary data to the function space valued PDE solution. They are also the natural mathematical framework for inverse problems, both in the context of PDEs [12] and in (medical) imaging [3], where the object of interest is the *inverse operator* which maps observables to the underlying material property/image that needs to be inferred or reconstructed.

Given the prohibitive cost of traditional physics based algorithms for approximating operators, particularly those associated with PDEs, increasing attention is being paid in recent years to machine-learning based *operator approximation*. Such techniques for learning operators from data fall under the genre of *operator learning*. A (by no means comprehensive) list of examples for operator learning architectures include operator networks [6], DeepONets [20], Graph neural operators [18], multipole neural operators [19], PCA-Nets [4], Fourier neural operators (FNO) [17], VIDON [21], spectral neural operators [10], LOCA [13], NOMAD [24], Continuous Generative Neural Networks [1] and transformer based operator learning models [5].

However, there is still a lack of clarity on what constitutes *operator learning?* Clearly, an operator learning framework, or a *neural operator* in the nomenclature of [15], should be able to *process functions as inputs and outputs*. On the other hand, functions are continuous objects and in practice,

37th Conference on Neural Information Processing Systems (NeurIPS 2023).

one may not have access to functions either at the input or output level. Instead, one can only access functions and perform computations with them on digital computers through their *discrete representations* such as point values on a grid, cell averages or in general, coefficients of an underlying basis. Hence, in practice, neural operators have to map discrete inputs to discrete outputs. This leads to a dichotomy, i.e., neural operators are designed to process functions as inputs/outputs but only have access to their discrete representations. Consequently, at any finite resolution, possible mismatches between the discretizations and the continuous versions of neural operators can lead to inconsistencies in the underlying function spaces. These inconsistency errors can propagate through the networks and may mar the performance of these algorithms. Moreover, important structural properties of the underlying operator, such as symmetries and conservation laws, hold at the continuous level. Inconsistent discretizations may not preserve these structural properties of the operator, leading to symmetry breaking etc. with attendant adverse consequences for operator approximation.

Addressing this possible inconsistency between neural operators and their discretizations is the central point of this article. To this end, we revisit and adapt existing notions of the relationship between continuous and discrete representations in signal processing, applied harmonic analysis and numerical analysis, with respect to the questions of sampling and interpolation. Roughly speaking, we aim to find a mathematical framework in which the continuous objects (functions) can be completely (uniquely and stably) recovered from their discrete representations (point evaluations, basis coefficients, etc.) at *any* resolution. Consequently, working with discrete values is tantamount to accessing the underlying continuous object. This equivalence between the continuous and the discrete is leveraged to define *a class of neural operators*, for which the discrete input-output representations or their continuous function space realizations are *equivalent*. On the other hand, lack of this equivalence leads to *aliasing errors*, which can propagate through the neural operator layers to adversely affect operator approximation. More concretely, our contributions are

- We provide a novel, very general unifying mathematical formalism to characterize a class of neural operators such that there is an equivalence between their continuous and discrete representations. These neural operators are termed as *Representation equivalent Neural Operators* or ReNOs. Our definition results in an automatically consistent function space formulation for ReNOs.

- To define ReNOs, we provide a novel and precise quantification of the notion of *aliasing error for operators*. Consequently, ReNOs are neural operators with zero aliasing error.

- We analyze existing operator learning architectures to find whether they are ReNOs or not.

- Synthetic numerical experiments are presented to illustrate ReNOs learn operators without aliasing and to also point out the practical consequences of aliasing errors particularly, with respect to evaluations of the neural operators on different grid resolutions.

## 2   A short background on Frame Theory and Aliasing

We start with a short discussion, revisiting concepts on the relationship between functions and their discrete representations. This is the very essence of *frame theory,* widely used in signal processing and applied harmonic analysis. We refer to SM D for a detailed introduction to frame theory.

**Equivalence between functions and their point samples.**   For simplicity of exposition, we start with univariate functions $f \in L^2(\mathbb{R})$ and the following question: Can a function be uniquely and stably recovered from its values, *sampled* from equispaced grid points $\{f(nT)\}_{n \in \mathbb{Z}}$ ? The classical *Whittaker-Shannon-Kotel'nikov (WSK) sampling theorem* [25], which lies at the heart of digital-to-analog conversion, answers this question in the affirmative when the underlying function $f \in \mathcal{B}_\Omega$, i.e., it belongs to the *Paley-Wiener space* of *band-limited functions* $\mathcal{B}_\Omega = \{f \in L^2(\mathbb{R}) : \operatorname{supp} \hat{f} \subseteq [-\Omega, \Omega]\}$, for some $\Omega > 0$, $\hat{f}$ the Fourier transform of $f$ and *sampling rate* $1/T \geq 2\Omega$, with $2\Omega$ termed as the *Nyquist rate*. The corresponding reconstruction formula is,

$$f(x) = 2T\Omega \sum_{n \in \mathbb{Z}} f(nT)\operatorname{sinc}(2\Omega(x - nT)), \quad \operatorname{sinc}(x) = \sin(\pi x)/(\pi x). \tag{2.1}$$

We note that the sequence of functions $\{\phi_n(x) = \mathrm{sinc}(2\Omega x - n)\}_{n \in \mathbb{Z}}$ constitutes an *orthonormal basis* for $\mathcal{B}_\Omega$ and denote by $\mathcal{P}_{\mathcal{B}_\Omega}\colon L^2(\mathbb{R}) \to \mathcal{B}_\Omega$ the *orthogonal projection operator* onto $\mathcal{B}_\Omega$,

$$\mathcal{P}_{\mathcal{B}_\Omega} f = \sum_{n \in \mathbb{Z}} \langle f, \phi_n \rangle \phi_n = \sum_{n \in \mathbb{Z}} f\left(\frac{n}{2\Omega}\right)\phi_n, \tag{2.2}$$

where the last equality is a consequence of $\mathcal{B}_\Omega$ being a reproducing kernel Hilbert space with kernel $\mathcal{K}_\Omega(x, y) = \mathrm{sinc}(2\Omega(x-y))$. Hence, $\mathcal{P}_{\mathcal{B}_\Omega} f$ corresponds to the right-hand side of (2.1) for $T = 1/2\Omega$ and formula (2.1) is exact if and only if $f \in \mathcal{B}_\Omega$, i.e., $f = \mathcal{P}_{\mathcal{B}_\Omega} f \iff f \in \mathcal{B}_\Omega$.

What happens when we sample a function at a sampling rate below the Nyquist rate, i.e. when $1/T < 2\Omega$? Alternatively, what is the effect of approximating a function $f \notin \mathcal{B}_\Omega$ by $\mathcal{P}_{\mathcal{B}_\Omega} f$? Again, sampling theory provides an answer in the form of the *aliasing error:*

**Definition 2.1. *Aliasing for bandlimited functions[2, §5]* ** *The* aliasing error function $\varepsilon(f)$ *and the corresponding* aliasing error *of $f \in L^2(\mathbb{R})$ for sampling at the rate $2\Omega$ are given by*

$$\varepsilon(f) = f - \mathcal{P}_{\mathcal{B}_\Omega} f, \qquad and \qquad \|\varepsilon(f)\|_2 = \|f - \mathcal{P}_{\mathcal{B}_\Omega} f\|_2.$$

*If the aliasing error $\varepsilon(f)$ is zero, i.e. if $f \in \mathcal{B}_\Omega$, we say that there is a* continuous-discrete equivalence (CDE) *between $f$ and its samples $\{f(n/2\Omega)\}_{n \in \mathbb{Z}}$.*

**Continuous-Discrete Equivalence in Hilbert spaces.**   Next, we generalize the above described concept of continuous-discrete equivalence to any separable Hilbert space $\mathcal{H}$, with inner product $\langle \cdot, \cdot \rangle$ and norm $\| \cdot \|$. A countable sequence of vectors $\{f_i\}_{i \in I}$ in $\mathcal{H}$ is a *frame* for $\mathcal{H}$ if there exist constants $A, B > 0$ such that for all $f \in \mathcal{H}$

$$A\|f\|^2 \le \sum_{i \in I} |\langle f, f_i \rangle|^2 \le B\|f\|^2. \tag{2.3}$$

Clearly, an orthonormal basis for $\mathcal{H}$ is an example of a frame with $A = B = 1$. We will now define maps that will allow us to make the link between functions and their discrete representations, which we will extensively use in the following sections. The bounded operator $T\colon \ell^2(I) \to \mathcal{H}$,  $T(\{c_i\}_{i \in I}) = \sum_{i \in I} c_i f_i$, is called *synthesis operator* and its adjoint $T^*\colon \mathcal{H} \to \ell^2(I)$,  $T^* f = \{\langle f, f_i \rangle\}_{i \in I}$, which discretizes the function by extracting its frame coefficients, is called *analysis operator*. By composing $T$ and $T^*$, we obtain the *frame operator* $S := TT^*$, which is an invertible, self-adjoint and positive operator. Furthermore, the pseudo-inverse of the synthesis operator is given by $T^\dagger\colon \mathcal{H} \to \ell^2(I)$,  $T^\dagger f = \{\langle f, S^{-1} f_i \rangle\}_{i \in I}$, which will be used in the following to reconstruct the function from its discrete representation, i.e. its frame coefficients.

With these concepts, one can introduce the most prominent result in frame theory [7], the *frame decomposition theorem*, which states that every element in $\mathcal{H}$ can be *uniquely and stably* reconstructed from its frame coefficients by means of the reconstruction formula

$$f = TT^\dagger f = \sum_{i \in I} \langle f, S^{-1} f_i \rangle f_i = \sum_{i \in I} \langle f, f_i \rangle S^{-1} f_i, \tag{2.4}$$

where the series converge unconditionally. Formula (2.4) is clearly a generalization of reconstruction formula (2.1). However, it is worth pointing out that, in general, the coefficients in (2.4) are not necessarily point samples of the underlying function $f$, but more general frame coefficients.

In general, it may not be possible to access all the frame coefficients to reconstruct a function in a Hilbert space. Instead, just like in the case of reconstructing functions from point samples, one will need to consider approximations to this idealized situation. This is best encapsulated by the notion of a *frame sequence*, i.e., a countable sequence $\{v_i\}_{i \in I} \in \mathcal{H}$ which is a frame for its closed linear span, i.e. for $\overline{\mathrm{span}}\{v_i : i \in I\}$. With this notion, we are in a position to generalize aliasing errors and the CDE, to arbitrary Hilbert spaces. Let $\mathcal{H}$ be a separable Hilbert space and let $\{v_i\}_{i \in I}$ be a frame sequence for $\mathcal{H}$ with $\mathcal{V} = \overline{\mathrm{span}}\{v_i : i \in I\}$ and frame operator $S\colon \mathcal{V} \to \mathcal{V}$. Then, the orthogonal projection of $\mathcal{H}$ onto $\mathcal{V}$ is given by $\mathcal{P}_\mathcal{V} f = \sum_{i \in I} \langle f, v_i \rangle S^{-1} v_i$. Thus, formula (2.4) holds and the function $f$ can be uniquely and stably recovered from its frame coefficients if and only if $f \in \mathcal{V}$. If $f \notin \mathcal{V}$, reconstructing $f$ from the corresponding frame coefficients results in an *aliasing error:*

**Definition 2.2. *Aliasing for functions in arbitrary separable Hilbert spaces.* ** *The* aliasing error function $\varepsilon(f)$ *and the resulting* aliasing error $\|\varepsilon(f)\|$ *of $f \in \mathcal{H}$ for the frame sequence $\{v_i\}_{i \in I} \subseteq \mathcal{V}$ are given by*

$$\varepsilon(f) = f - \mathcal{P}_\mathcal{V} f, \quad \|\varepsilon(f)\| = \|f - \mathcal{P}_\mathcal{V} f\|.$$

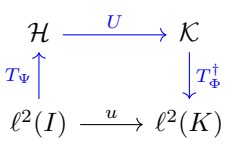

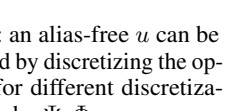

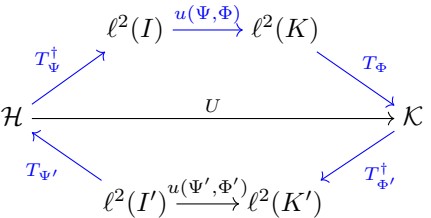

(a) An alias-free operator's diagram commutes.

(b) *ReNO*: an alias-free $u$ can be constructed by discretizing the operator $U$ for different discretizations given by $\Psi, \Phi$.

(c) *ReNO*: discrete representations $u, u'$ for different discretizations are equivalent.

Figure 1: Alias-free framework. $U$ is the underlying operator, $u$ its discrete implementation. The synthesis operators and their pseudo-inverses make the link between function and discrete space.

If the aliasing error $\varepsilon(f)$ is zero, i.e. if $f \in \mathcal{V}$, we say that there is a *continuous-discrete equivalence (CDE)* between $f$ and its frame coefficients $\{\langle f, v_i \rangle\}_{i \in I}$.

## 3 Alias-Free Framework for Operator Learning

In this section, we will extend the concept of aliasing of functions to operators. We demonstrate that this notion of *operator aliasing is fundamental to understanding how the neural operator performs across various discretizations*, thereby addressing the shortcomings of possible lack of *continuous-discrete equivalence*. We will then define Representation equivalent Neural operators (ReNOs), whose discrete representations are equivalent across discretizations.

Assume we posit a (neural) operator $U$, mapping between infinite-dimensional function spaces. As mentioned before, this operator is never computed in practice, instead a discrete mapping $u$ is used. The difficulty in formalizing this is that we should be able to compute this discrete mapping for any discretizations of the input and output function. We formalize this notion below in a practical manner.

**Setting.** Let $U \colon \operatorname{Dom} U \subseteq \mathcal{H} \to \mathcal{K}$ be an operator between two separable Hilbert spaces, and let $\Psi = \{\psi_i\}_{i \in I}$ and $\Phi = \{\phi_k\}_{k \in K}$ be frame sequences for $\mathcal{H}$ and $\mathcal{K}$, respectively, with synthesis operators $T_\Psi$ and $T_\Phi$. We denote their closed linear spans by $\mathcal{M}_\Psi := \overline{\operatorname{span}}\{\psi_i : i \in I\}$ and $\mathcal{M}_\Phi := \overline{\operatorname{span}}\{\phi_k : k \in K\}$. We note that by classical frame theory [7], the pseudo-inverses $T_\Psi^\dagger$ and $T_\Phi^\dagger$, initially defined on $\mathcal{M}_\Psi$ and $\mathcal{M}_\Phi$, respectively, can in fact be extended to the entire Hilbert spaces, i.e. $T_\Psi^\dagger : \mathcal{H} \to \ell^2(I)$ and $T_\Phi^\dagger : \mathcal{K} \to \ell^2(K)$.

### 3.1 Operator Aliasing and Representation Equivalence

Once the discretization is chosen – determined by input and output frame sequences $(\Psi, \Phi)$ – connecting the continuous operator $U$ with its discrete counterpart $u$ is the notion of operator aliasing. Given any mapping $u \colon \ell^2(I) \to \ell^2(K)$, we can build the operator $T_\Phi \circ u \circ T_\Psi^\dagger \colon \mathcal{H} \to \mathcal{K}$, whose definition clearly depends on the choices of the frame sequences that we make on the continuous level. In other words, any mapping $u$ can be interpreted as a discrete representation of an underlying continuous operator, which in general, may differ from the operator $U$, that is of interest here. Hence, in analogy to Definitions 2.1 and 2.2, we can define the aliasing error of $U$ relative to the discrete representation $u$ as,

**Definition 3.1.** *Operator aliasing.* *The aliasing error operator $\varepsilon(U, u, \Psi, \Phi) \colon \operatorname{Dom} U \subseteq \mathcal{H} \to \mathcal{K}$ is given by*

$$\varepsilon(U, u, \Psi, \Phi) = U - T_\Phi \circ u \circ T_\Psi^\dagger,$$

*and the corresponding scalar error is $\|\varepsilon(U, u, \Psi, \Phi)\|$, with $\|\cdot\|$ denoting the operator norm.*

An aliasing error of zero implies that the operator $U$ can be perfectly represented by first discretizing the function with $T_\Psi^\dagger$, applying $u$, and then reconstructing with $T_\Phi$, or equivalently, that the diagram in Figure 1a commutes, i.e. the black and the blue directed paths in the diagram lead to the same result. If the aliasing error is zero, we say that $(U, u, \Psi, \Phi)$ satisfies a *continuous-discrete equivalence*

*(CDE),* implying that accessing the discrete representation $u$ is exactly the same as accessing the underlying continuous operator $U$.

**Example 3.2.** ***Aliasing with the magnitude squared operator.*** Consider the operator $U(f) = |f|^2$ as an operator from $\mathcal{B}_\Omega$ into $\mathcal{B}_{2\Omega}$. The choice to discretize inputs and outputs on the same grid $\left\{\frac{n}{2\Omega}\right\}_{n\in\mathbb{Z}}$ corresponds to choosing $\Psi = \Phi = \{\mathrm{sinc}(2\Omega x - n)\}_{n\in\mathbb{Z}}$, and to defining the discrete mapping $u\colon \ell^2(\mathbb{Z}) \to \ell^2(\mathbb{Z})$ by $u(v) = T_\Psi^\dagger \circ U \circ T_\Psi(v) = v \odot \overline{v}$, where $\odot$ denotes the entrywise product. Then, for every $f \in \mathcal{B}_\Omega$ such that $U(f) \in \mathcal{B}_{2\Omega} \setminus \mathcal{B}_\Omega$, we have

$$\varepsilon(U, u, \Psi, \Phi)(f) = U(f) - T_\Phi \circ T_\Phi^\dagger(U(f)) = U(f) - \mathcal{P}_{\mathcal{B}_\Omega}(U(f)) \neq 0,$$

and we encounter aliasing. This occurs because $U$ introduces new frequencies that exceed the bandwidth $\Omega$. However, we can rectify this by sampling the output functions on a grid with twice the resolution of the input grid. This corresponds to choosing $\Phi = \{\mathrm{sinc}(4\Omega x - n)\}_{n\in\mathbb{Z}}$ and to defining $u = T_\Phi^\dagger \circ U \circ T_\Psi$, which simply maps samples from grid points $\left\{\frac{n}{2\Omega}\right\}_{n\in\mathbb{Z}}$ into squared samples from the double resolution grid $\left\{\frac{n}{4\Omega}\right\}_{n\in\mathbb{Z}}$. This effectively removes aliasing since the equality $U = T_\Phi \circ u \circ T_\Psi^\dagger$ is satisfied. Furthermore, sampling the input and output functions with arbitrarily higher sampling rate, i.e. representing the functions with respect to the system $\{\mathrm{sinc}(2\overline{\Omega}x - n)\}_{n\in\mathbb{Z}}$ with $\overline{\Omega} > 2\Omega$, yields no aliasing error since $\{\mathrm{sinc}(2\overline{\Omega}x - n)\}_{n\in\mathbb{Z}}$ constitutes a frame for $\mathcal{B}_{2\Omega} \supseteq \mathcal{B}_\Omega$.

In practice, the discrete representation $u$ of the operator $U$ depends on the choice of the frame sequences. This means that – as mentioned at the beginning of the section – there is one map for every input/output frame sequence $\Psi, \Phi$. The consistency between operations $u, u'$ with respect to different frame sequences can be evaluated using the following error.

**Definition 3.3.** ***Representation equivalence error.*** *Suppose that $u, u'$ are discrete maps with associated frame sequences $\Psi, \Phi$ and $\Psi', \Phi'$, respectively. Then, the representation equivalence error is given by the function $\tau(u, u')\colon \ell^2(I) \to \ell^2(K)$, defined as:*

$$\tau(u, u') = u - T_\Phi^\dagger \circ T_{\Phi'} \circ u' \circ T_{\Psi'}^\dagger \circ T_\Psi$$

*and the corresponding scalar error is $\|\tau(u, u')\|$.*

Intuitively, this amounts to computing each mapping on their given discretization, and comparing them by expressing $u'$ in the frames associated to $u$. In the following section we leverage the notion of operator aliasing in the context of operator learning, as well as explore its practical implications with respect to representation equivalence at the discrete level.

## 3.2 Representation equivalent Neural Operators (ReNO)

Equipped with the above notion of continuous-discrete equivalence, we can now introduce the concept of *Representation equivalent Neural Operator (ReNO)*. To this end, for any pair $(\Psi, \Phi)$ of frame sequences for $\mathcal{H}$ and $\mathcal{K}$, we consider a mapping at the discrete level $u(\Psi, \Phi)\colon \mathrm{Ran}\, T_\Psi^\dagger \to \mathrm{Ran}\, T_\Phi^\dagger$, which handles discrete representations of the functions. Notice how this map is indexed by the frame sequences: this is normal as when the discretization changes, the definition of the function should also change. In order to alleviate the notation, when this is clear from the context, we will refer to $u(\Psi, \Phi)$ simply as $u$. See **SM** D.1 for an explanation of the condition $u(\Psi, \Phi)\colon \mathrm{Ran}\, T_\Psi^\dagger \to \mathrm{Ran}\, T_\Phi^\dagger$.

**Definition 3.4.** ***Representation equivalent Neural Operators (ReNO).*** *We say that $(U, u)$ is a ReNO if for every pair $(\Psi, \Phi)$ of frame sequences that satisfy $\mathrm{Dom}\, U \subseteq \mathcal{M}_\Psi$ and $\mathrm{Ran}\, U \subseteq \mathcal{M}_\Phi$ there is no aliasing, i.e. the aliasing error operator is identical to zero:*

$$\varepsilon(U, u, \Psi, \Phi) = 0. \tag{3.1}$$

*We will write this property in short as $\varepsilon(U, u) = 0$.*

In other words, the diagram in Figure 1a commutes for every considered pair $(\Psi, \Phi)$. In this case, the discrete representations $u(\Psi, \Phi)$ are all equivalent, meaning that they uniquely determine the same underlying operator $U$, whenever a continuous-discrete equivalence property holds at the level of the function spaces. The domain and range conditions in Definition 3.4 simply imply that the frames can adequately represent input and output functions of $U$.

**Remark 3.5.** If the aliasing error $\varepsilon(U, u, \Psi, \Phi)$ is zero (as required in Definition 3.4), then the assumption that $u(\Psi, \Phi)$ maps $\operatorname{Ran} T_\Psi^\dagger \subseteq \ell^2(I)$ into $\operatorname{Ran} T_\Phi^\dagger \subseteq \ell^2(K)$ implies that

$$u(\Psi, \Phi) = T_\Phi^\dagger \circ U \circ T_\Psi. \tag{3.2}$$

We observe that this definition of $u(\Psi, \Phi)$ is such that the diagram in Figure 1b commutes. In other words, once we fix the discrete representations $\Psi, \Phi$ associated to the input and output functions, there exists a unique way to define a discretization $u(\Psi, \Phi)$ that is consistent with the continuous operator $U$ and this is given by (3.2). In practice, we may have access to different discrete representations of the input and output functions, which in the theory amounts to a change of reference systems in the function spaces. Note that to avoid any aliasing error, the discrete representation of $U$ *has to* depend on the chosen frame sequences, i.e. inevitably, $u$ must depend on $\Psi$ and $\Phi$, and hence, must be discretization dependent. See A.2 for the proof of Remark 3.5.

In particular, Remark 3.5 directly implies a formula to go from one discrete representation to another, as

$$u(\Psi', \Phi') = T_{\Phi'}^\dagger \circ T_\Phi \circ u(\Psi, \Phi) \circ T_\Psi^\dagger \circ T_{\Psi'}, \tag{3.3}$$

whenever the pairs of frame sequences $(\Psi, \Phi)$ and $(\Psi', \Phi')$ satisfy the conditions in Definition 3.4. In other words, the diagram in 1c commutes.

Formula (3.3) immediately implies Proposition 3.6, which establishes a link between aliasing and representation equivalence. This highlights the contrast to *discretization invariance,* discussed at length in [15]: while this concept establishes an asymptotic consistency, representation equivalence includes the direct comparison between any two given discretizations and guarantees their equivalence.

**Proposition 3.6.** *Equivalence of ReNO discrete representations.* Let $(U, u)$ be a ReNO. For any two frame sequence pairs $(\Psi, \Phi)$ and $(\Psi', \Phi')$ satisfying conditions in Definition 3.4, we have that

$$\tau(u, u') = 0,$$

where, by a slight abuse of notation, $u'$ denotes $u(\Psi', \Phi')$.

Hence, under the assumption that the discrete map at each discretization is consistent with the underlying continuous operator, we have a unique way to express the operator at each discretization. Moreover, formula (3.3) closely resembles analogous formulas presented in [17, 15, 23] when evaluating *single shot super resolution*. However, in Section 5, we offer a nuanced perspective, indicating variability across different scenarios.

### 3.3  Layer-wise Instantiation

As shown in Remark 3.5, we can compute the outputs of the ReNO on the computer by first discretizing the continuous operator $U$. Typically, if $U$ is a neural operator composed of multiple layers, as we will show in this section, it is possible to discretize each layer, while remaining consistent with the underlying operator.

Consider a neural operator $U$ with $L$ layers, each a mapping between separable Hilbert spaces:

$$U = U_L \circ U_{L-1} \circ \ldots \circ U_1, \qquad U_\ell \colon \mathcal{H}_\ell \to \mathcal{H}_{\ell+1}, \quad \ell = 1, \ldots, L, \tag{3.4}$$

we denote by $\Psi_\ell$ a *frame sequence* for $\mathcal{H}_\ell$. The choice of frame sequences for each $\mathcal{H}_\ell$ corresponds to the choice of accessible discrete representations of functions in the underlying function space $\mathcal{H}_\ell$.

**Proposition 3.7.** *Stability of ReNO under composition.* *Consider the composition $U = U_L \circ \ldots \circ U_1$ as in eq. 3.4, as well as a discrete mapping $u = u_L \circ \ldots \circ u_1$. If each layer $(U_\ell, u_\ell)$ is a ReNO, the composition $(U, u)$ also is a ReNO.*

As the proof of Proposition 3.7, presented in **SM** A.1, also shows, if each hidden layer in the operator (3.4) has an aliasing error (2.2), then these errors may propagate through the network and increase with increasing number of layers.

### 3.4  $\epsilon-$**ReNOs**

In practice, we may not need to set the aliasing error to zero. But it is essential to be able to control it and make it as small as desired. The notion of ReNOs, Definition 3.4, can very simply be extended to the case where we allow for a small, controlled amount of aliasing. Indeed we can introduce $\epsilon-$ReNOs, which satisfy $\|\varepsilon(U, u)\| \leq \epsilon$ (for every pair of admissible frame sequences) instead of $\varepsilon(U, u) = 0$.

**Proposition 3.8.** *Let $(U, u)$ be an $\epsilon-$ReNO. For any two frame sequence pairs $(\Psi, \Phi)$ and $(\Psi', \Phi')$ satisfying conditions in Definition 3.4 and such that $\mathcal{M}_{\Phi'} \subseteq \mathcal{M}_\Phi$, we have*

$$\|\tau(u, u')\| \leq \frac{2\epsilon\sqrt{B_\Psi}}{\sqrt{A_\Phi}},$$

*where, by a slight abuse of notation, $u'$ denotes $u(\Psi', \Phi')$.*

## 4  Examples

Equipped with our definition of Representation equivalent Neural Operators (ReNOs), we analyze some existing operator learning architectures to ascertain whether they are neural operators or not.

**Convolutional Neural Networks (CNN).**   Classical convolutional neural networks are based on the convolutional layer $k$, involving a discrete kernel $f \in \mathbb{R}^{2s+1}$ and a discrete input $c$:

$$k(c)[m] = (f * c)[m] = \sum_{i=-s}^{s} c[m-i]f[i].$$

We can then analyze this layer using our framework to ask whether this operation can be associated to some underlying continuous operator. Intuitively, if this is the case, the computations conducted on different discretizations effectively representing the input should be consistent; in the contrary case, no associated continuous operator exists and the convolutional operation is not a ReNO.

Consider the case where the discrete input $c$ corresponds to pointwise evaluation on a grid of some underlying bandlimited function $f \in \mathcal{B}_\Omega$, for example $c[n] = f\left(\frac{n}{2\Omega}\right), n \in \mathbb{Z}$ with associated orthonormal basis $\Psi = \{\mathrm{sinc}(2\Omega x - n)\}_{n \in \mathbb{Z}}$. Consider now an alternate representation of $f$, point samples of a grid twice as fine: $d[n] = f\left(\frac{n}{4\Omega}\right)$, with $\Psi' = \{\mathrm{sinc}(4\Omega x - n)\}_{n \in \mathbb{Z}}$ as basis. Clearly, even though discrete inputs agree, i.e. $c[n] = d[2n]$, this is no longer true for the outputs, $(k * c)[n] \neq (k * d)[2n]$. This in turn implies that there exist frame sequences $\Psi, \Psi'$ such that:

$$T_\Psi \circ k_\Psi(c) \circ T_\Psi^\dagger \neq T_{\Psi'} \circ k_{\Psi'}(d) \circ T_{\Psi'}^\dagger, \tag{4.1}$$

thereby defying the representation equivalence property of ReNOs, in the sense of Definition 3.4. This fact is also corroborated with the experimental analysis, Section 5.

**Fourier Neural Operators (FNO).**   FNOs are defined in [17] in terms of layers, which are either lifting or projection layers or *Fourier layers*. As lifting and projection layers do not change the underlying spatial structure of the input function, but only act on the channel width, these linear layers will satisfy Definition 3.4 of ReNO here. Hence, we focus on the Fourier layer of the form,

$$v_{\ell+1}(x) = \sigma\left(A_\ell v_\ell(x) + B_\ell(x) + \mathcal{K}v_\ell(x)\right), \tag{4.2}$$

with the Fourier operator given by

$$\mathcal{K}v = \mathcal{F}^{-1}(R \odot \mathcal{F})(v).$$

Here, $\mathcal{F}, \mathcal{F}^{-1}$ are the Fourier and Inverse Fourier transforms.

For simplicity of exposition, we set $A_\ell = B_\ell \equiv 0$ and focus on investigating whether the Fourier layer (4.2) satisfies the requirements of a ReNO. Following [17], the discrete form of the Fourier layer is given by $\sigma(kv)$, with $kv = F^{-1}(R \odot F(v))$, where $F, F^{-1}$ denote the discrete Fourier transform (DFT) and its inverse.

In **SM** B.1, we show that the convolution in Fourier space operation for the FNO layer (4.2) satisfies the requirements of a ReNO. However the pointwise activation function $\sigma(f)$, applied to a bandlimited input $f \in \mathcal{B}_\Omega$ will not necessarily respect the bandlimits, i.e., $\sigma(f) \notin \mathcal{B}_\Omega$. In fact, with popular choices of activation functions such as ReLU, $\sigma(f) \notin \mathcal{B}_\omega$, for any $\omega > 0$ (see **SM** C for numerical illustrations). Thus, the Fourier layer operator (4.2) may not respect the continuous-discrete equivalence and can lead to aliasing errors, a fact already identified in [10]. Hence, FNO *may not be a ReNO in the sense of Definition 3.4.*

**Convolutional Neural Operators (CNO).** Introduced in [23], the layers of a convolutional neural operator consist of three elementary operations

$$v_{l+1} = \mathcal{P}_l \circ \Sigma_l \circ \mathcal{K}_l(v_l), \quad 0 \le l \le L - 1, \tag{4.3}$$

where $\mathcal{K}_l$ is a convolution operator, $\Sigma_l$ is a non-linear operator whose definition depends on the choice of an activation function $\sigma \colon \mathbb{R} \to \mathbb{R}$, and $\mathcal{P}_l$ is a projection operator. We show in **SM** B.2 that CNO layers respect equation (3.2) and consequently CNOs are Representation equivalent Neural Operators (ReNOs) in the sense of Definition 3.4. This is the result of the fact that the activation layer is defined in a way to respect the band-limits of the underlying function space.

## 5 Empirical Analysis

### 5.1 Assessing Representation Equivalence

In the previous section, we have studied existing neural operator architectures from a theoretical perspective. In this section, for the same architectures, we corroborate these findings from an empirical viewpoint. Aliasing is a quantity that cannot be computed in practice, as we cannot access the underlying operator $U$ on a computer, we can nonetheless compute representation equivalent errors introduced in Definition 3.3, which is related to aliasing by Propositions 3.7 and 3.8 (as well as being a quantity of interest in itself).

**Experimental Setting.** We wish to learn an unknown target operator $Q$ using a neural operator. In this experiment, all neural operators (CNN, FNO and SNO) take as input pointwise evaluations on the grid, and are able to deal with varying input resolutions. A simple way of constructing the target operator $Q : H \to H$, where $H$ is the space of periodic and $K = 30$-bandlimited functions, is by sampling input and output pairs in a random fashion. Sampling of a function in $H$ can be realized as follows: As we know that $\Psi_K := \{d_K(. - x_k)\}_{k=-K,\dots,K}$ constitutes a frame for $H$, $d_K$ being the Dirichlet kernel of order $K$ and $x_k = \frac{k}{2K+1}$, any function $f \in H$ can be written as $f(x) = \sum_{k=-K}^{K} f(x_k) d_K(x - x_k)$. Thus, the discrete representation of $f$ simply corresponds to its $2K + 1 = 61$ point-wise evaluations on a grid, i.e. $\{f(x_k)\}_{k=-K,\dots,K}$. Note that for simplicity we have used and will use the same frame sequences for both input and output spaces, as these are the same.

**Training and Evaluation.** Once the data is generated, we train neural operators $u(\Psi_K, \Psi_K)$ on discretizations associated to the frame sequences, which are simply the point-wise evaluations of the input-target functions in the data. In other words, we regress to the frame coefficients of the target function, with coefficients of the input function as input. Once training is over, we evaluate how the different neural operators behave when dealing with changing input and output frame sequences. The frame sequences here are $\Psi_M$ for different testing frames, with associated $2M + 1$ sized grids, with associated operator $u_M : \mathbb{R}^{2M+1} \to \mathbb{R}^{2M+1}$.

The results of this experiment are presented in Figure 2; we also provide results for additional architectures in Figure 4. The results in Figure 2 clearly show that as predicted by our theory, neither CNN nor FNO are representation equivalent, in the sense of Definition 3.4 and changing the resolution, which amounts to a change of frame, does not keep the operator invariant, causing aliasing errors that materialize themselves as representation equivalence errors (as in Definition 3.3) here. On the other hand, as predicted by the theory, CNO is more likely to be a ReNO – as long as the frames selected satisfy conditions of Definition 3.4 (i.e. "Representation Equivalence" zone). When these conditions no longer hold ("No Equivalence" zone), CNO also generates aliasing errors. Thus, this experiment clearly demonstrates the practical implications of the ReNO framework. In contrast, the

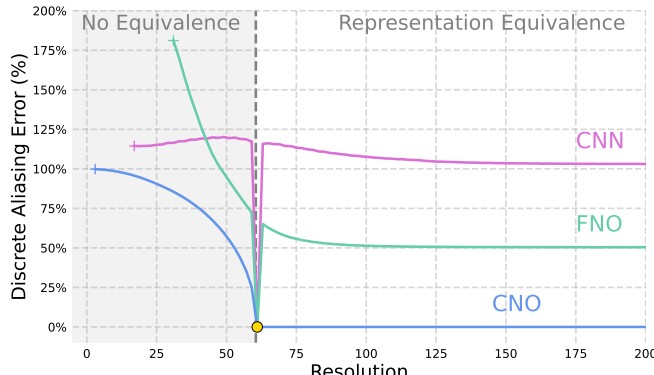

Figure 2: Representation equivalence is computed for three classical architectures, CNN, FNO and CNO trained on a resolution of 61 (yellow dot). The "Resolution Equivalence" zone, located on the right-hand side, denotes the region where discrete representations have an associated frame, while the left-hand side represents the area where this is no longer the case (loss of input/output information). As predicted by the theory, CNN and FNO are not representation equivalent, while CNO is error-free in the equivalence region, but failing to do so outside of it.

*discretization invariance* [15] condition only ensures that in the infinite resolution limit, the neural operator converges, while not making any predictions about the behavior of the neural operator at finite resolutions.

**Minimizing representation equivalence error.**    Instead of directly tackling aliasing errors from a theoretical perspective as is done in the previous sections, we select an architecture that may have aliasing, i.e. FNO, and we try to minimize the representation equivalence error during training. We observe and detail our findings in **SM** C.3.

## 5.2   Assessing Structure Preservation

| | | | | |
|---|---|---|---|---|
| Representation Equivalence Error (%) | 10.38 | 20.24 | 42.97 | 98.55 |
| Translation Equivariance Error (%) | 10.54 | 20.54 | 45.79 | 104.44 |

Table 1: Representation equivalence error vs translation equivariance error. We observe a direct link between aliasing and preserving continuous structures.

Our methodology aims to maintain the structure of the underlying operator, as defined at the continuous level. Fundamental structural characteristics of the underlying operator, like symmetries and conservation principles, are maintained at the discrete level. Discretizations that are not aligned may fail to uphold these intrinsic properties of the operator, resulting in deviations like symmetry breaking which can negatively influence the results.

As an example of this, we consider FNO. Similarly to what is done in [11], the primary structure we aim to uphold is translation equivariance, as defined at the operator level. We look at different trained FNOs – all of which have a different representation equivalence error – and assess whether they are translation equivariant or not. More specifically, these FNOs are trained by minimizing both the regression loss and the representation equivalence error simultaneously, introducing a more or less large multiplier $\lambda$ on the latter, just like in **SM** C.3.

As we vary the parameter $\lambda$, we observe in Table 1, there is a distinct relationship between the Discrete Aliasing Error (DAE) and the Translation Equivariance Error. Both errors show a similar trend with a nearly perfect positive linear association.

## 6 Discussion

**Summary.** Although a variety of architectures for operator learning are available since the last couple of years, there is still a lack of clarity about what exactly constitutes *operator learning*. Everyone would agree that an operator learning architecture should have functions as inputs and outputs. However, in practice, one does not necessarily have access to functions, either at the input or output level. Rather the access is limited to some form of *discrete representations* of the underlying functions, for instance, point values, cell averages, coefficients with respect to a basis etc. Moreover, one can only perform computations with discrete objects on digital computers. Hence, given the necessity of having to work with discrete representations of functions, it is essential to enforce some relationship between the continuous and discrete representations of the underlying functions or in other words, demand consistency in function space.

Unlike in [15], where the authors advanced a form of *asymptotic consistency* in terms of discretization parameters, we go further to require a stronger form of structure-preserving equivalence of the continuous and discrete representations of operators. To this end, we leverage the equivalence between continuous and discrete representations of functions in Hilbert spaces in terms of frame theory. The main point about this equivalence is the fact that functions, belonging to suitable function spaces, can be uniquely and stably reconstructed, from their frame coefficients. A failure to enforce this equivalence results in the so-called aliasing errors that quantitatively measure function space inconsistencies.

We extend this notion of aliasing error to operators here and use it to define *Representation equivalent Neural Operators (ReNOs)*, see Definition 3.4. Our framework automatically implies consistency in function spaces and provides a recipe for deriving ReNOs in terms of changing the underlying frames. We also employ our framework to analyze whether or not existing operator learning architectures are ReNOs and also corroborate our results through experiments.

**Related Work.** Our current paper relies heavily on structure preservation from classical numerical analysis, [22] and references therein, as well as concepts from signal processing and applied harmonic analysis [25] and references therein. Among the emerging literature on operator learning, [15] was one of the first papers to attempt a unifying framework that encompasses many operator learning architectures and codify them through a particular definition of neural operators. We take an analogous route here but with our main point of departure from [15] being that unlike their notion of asymptotic consistency, we require *systematic consistency in function space* by enforcing the representation equivalence for the underlying operator at each layer. Another relevant work is [10] where the authors flag the issue of possible aliasing errors with specific operator learning architectures. We significantly expand on the approach of [10] by providing a rigorous and very general definition for aliasing errors. Finally, our definition of Representation equivalent Neural Operators, relying on aliasing errors for operators, is analogous to a similar approach for *computing with functions*, rather than discrete representations, which is the cornerstone of the *Chebfun* project [8] and references therein.

**Limitations and Extensions.** Our aim in this paper was to tackle a fundamental question of what defines a *neural operator?* We have addressed this question here and shown that enforcing some form of equivalence between continuous and discrete representations is needed for the architecture to genuinely learn the underlying operator rather than just a discrete representation of it. What we have not addressed here are quantitative measures of the error associated with a *Representation equivalent Neural Operator,* introduced in Definition 3.4, in approximating the underlying operator. This is a much broader question than what is addressed here, since sources of error, other than aliasing errors, such as *approximation, training and generalization* errors also contribute to the total error (see [16, 14, 15]).

One interesting direction for further analysis would involve exploring operators and their discretized counterparts that form $\epsilon$-ReNOs and also enforce the ReNO property approximately. Characterizing the implications of this concept within operator learning is a topic for future work.

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
