## A  Proofs

### A.1  Proof of Proposition 3.7

We show the proof of Proposition 3.7 for a two layer Representation equivalent Neural Operator. The proof generalizes to the case of $\ell > 2$ layers. We want to show that the aliasing error $\varepsilon(U_2 \circ U_1, u_2 \circ u_1, \Psi_1, \Psi_3)$ is identically zero whenever the frame sequences satisfy the conditions in Definition 3.4, or equivalently whenever the diagram

$$
\begin{array}{ccccc}
\mathcal{H}_1 & \xrightarrow{\;\;U_1\;\;} & \mathcal{H}_2 & \xrightarrow{\;\;U_2\;\;} & \mathcal{H}_3 \\
\downarrow{\scriptstyle T_{\Psi_1}^{\dagger}} & {\scriptstyle T_{\Psi_2}}\;\uparrow\downarrow{\scriptstyle T_{\Psi_2}^{\dagger}} & & {\scriptstyle T_{\Psi_3}}\;\uparrow & \\
\ell^2(I_1) & \xrightarrow{u_1(\Psi_1,\Psi_2)} & \ell^2(I_2) & \xrightarrow{u_2(\Psi_2,\Psi_3)} & \ell^2(I_3)
\end{array}
$$

is commutative. We directly compute

$$
\begin{aligned}
U_2 \circ U_1 &= (T_{\Psi_3} \circ u_2(\Psi_2, \Psi_3) \circ T_{\Psi_2}^{\dagger}) \circ (T_{\Psi_2} \circ u_1(\Psi_1, \Psi_2) \circ T_{\Psi_1}^{\dagger}) \\
&= T_{\Psi_3} \circ u_2(\Psi_2, \Psi_3) \circ u_1(\Psi_1, \Psi_2) \circ T_{\Psi_1}^{\dagger}.
\end{aligned}
$$

The first equality simply follows by the definition of ReNO. The last equality can be seen as follows: first, $\mathcal{M}_{\Psi_2} = \mathrm{Ran}(T_{\Psi_2} T_{\Psi_2}^{*}) \subseteq \mathrm{Ran}(T_{\Psi_2}) \subseteq \mathcal{M}_{\Psi_2}$, so that $\mathrm{Ran}(T_{\Psi_2})$ is closed. This, in combination with Lemma 2.5.2 in [7], implies that $\mathrm{Ran}(T_{\Psi_2}^{\dagger})$ is also closed. Hence, $T_{\Psi_2}^{\dagger} \circ T_{\Psi_2}$ is the orthogonal projection onto $(\mathrm{Ker}(T_{\Psi_2}))^{\perp} = \mathrm{Ran}(T_{\Psi_2}^{\dagger})$ and, by assumption, $\mathrm{Ran}(u_1(\Psi_1, \Psi_2) \circ T_{\Psi_1}^{\dagger}) \subseteq \mathrm{Ran}(T_{\Psi_2}^{\dagger})$. As a consequence, the aliasing error operator (3.1) of $U_2 \circ U_1$ for the discretized version $u_2(\Psi_2, \Psi_3) \circ u_1(\Psi_1, \Psi_2)$ is identically zero, which proves the hypothesis.

### A.2  Proof of Remark 3.5

We keep the notation as in Section 3.2. If the aliasing error $\varepsilon(U, u, \Psi, \Phi)$ is zero, then

$$
U = T_{\Phi} \circ u(\Psi, \Phi) \circ T_{\Psi}^{\dagger}. \tag{A.1}
$$

By equation (A.1) we readily obtain

$$
T_{\Phi}^{\dagger} \circ U \circ T_{\Psi} = T_{\Phi}^{\dagger} \circ T_{\Phi} \circ u(\Psi, \Phi) \circ T_{\Psi}^{\dagger} \circ T_{\Psi} = u(\Psi, \Phi),
$$

where the last equality follows by the fact that $T_{\Psi}^{\dagger} \circ T_{\Psi}$ is the orthogonal projection onto $(\mathrm{Ker}(T_{\Psi}))^{\perp} = \mathrm{Ran}(T_{\Psi}^{\dagger})$ and, by assumption, $u(\Psi, \Phi)$ maps $\mathrm{Ran}\, T_{\Psi}^{\dagger}$ into $\mathrm{Ran}\, T_{\Phi}^{\dagger}$. This concludes the proof of Remark 3.5.

### A.3  Proof of Proposition 3.8

Let $(\Psi, \Phi)$ and $(\Psi', \Phi')$ be two frame sequence pairs satisfying conditions in 3.4, and let $A_{\Phi}$ be the lower frame bound of $\Phi$ and $B_{\Psi}$ the upper frame bound of $\Psi$. A standard result in frame theory is that $T_{\Phi}^{\dagger}$ is a bounded operator on $\mathcal{M}_{\Phi}$ with $\|T_{\Phi}^{\dagger}\| \leq 1/\sqrt{A_{\Phi}}$. Also, the norm of the synthesis operator is bounded by the upper frame bound, resulting in $\|T_{\Psi}\| \leq \sqrt{B_{\Psi}}$. By assumption, we know that

$$
\|U - T_{\Phi} \circ u(\Psi, \Phi) \circ T_{\Psi}^{\dagger}\| \leq \epsilon, \qquad \|U - T_{\Phi'} \circ u(\Psi', \Phi') \circ T_{\Psi'}^{\dagger}\| \leq \epsilon.
$$

The representation equivalence error reads

$$
\tau(u(\Psi, \Phi), u(\Psi', \Phi')) = u(\Psi, \Phi) - T_{\Phi}^{\dagger} \circ T_{\Phi'} \circ u(\Psi', \Phi') \circ T_{\Psi'}^{\dagger} \circ T_{\Psi}.
$$

Therefore, employing the linearity of the synthesis operators and their pseudo-inverses, we can estimate

$$\|\tau(u(\Psi,\Phi),u(\Psi',\Phi'))\|$$

$$\leq \|u(\Psi,\Phi) - T_\Phi^\dagger \circ U \circ T_\Psi\| + \|T_\Phi^\dagger \circ U \circ T_\Psi - T_\Phi^\dagger \circ T_{\Phi'} \circ u(\Psi',\Phi') \circ T_{\Psi'}^\dagger \circ T_\Psi\|$$

$$\leq \|u(\Psi,\Phi) - T_\Phi^\dagger \circ U \circ T_\Psi\| + \|T_\Phi^\dagger\|\|U - T_{\Phi'} \circ u(\Psi',\Phi') \circ T_{\Psi'}^\dagger\|\|T_\Psi\|$$

$$\leq \|u(\Psi,\Phi) - T_\Phi^\dagger \circ U \circ T_\Psi\| + \frac{\epsilon\sqrt{B_\Psi}}{\sqrt{A_\Phi}}.$$

In the above, we have used the fact that $T_\Phi^\dagger$ acts on $U - T_{\Phi'} \circ u(\Psi',\Phi') \circ T_{\Psi'}^\dagger$, which is an operator with range in $\operatorname{Ran} U \cup \mathcal{M}_{\Phi'} \subseteq \mathcal{M}_\Phi$, on which $T_\Phi^\dagger$ is bounded with $\|T_\Phi^\dagger\| \leq 1/\sqrt{A_\Phi}$.

We observe that if $u(\Psi,\Phi)\colon \operatorname{Ran}(T_\Psi^\dagger) \to \operatorname{Ran}(T_\Phi^\dagger)$, then

$$u(\Psi,\Phi) = T_\Phi^\dagger \circ T_\Phi \circ u(\Psi,\Phi) \circ T_\Psi^\dagger \circ T_\Psi,$$

where the equality follows by the fact that $T_\Phi^\dagger \circ T_\Phi$ and $T_\Psi^\dagger \circ T_\Psi$ are respectively the orthogonal projections onto $(\operatorname{Ker}(T_\Phi))^\perp = \operatorname{Ran}(T_\Phi^\dagger)$ and $(\operatorname{Ker}(T_\Psi))^\perp = \operatorname{Ran}(T_\Psi^\dagger)$. As a consequence,

$$\|\tau(u(\Psi,\Phi),u(\Psi',\Phi'))\| \leq \|T_\Phi^\dagger \circ T_\Phi \circ u(\Psi,\Phi) \circ T_\Psi^\dagger \circ T_\Psi - T_\Phi^\dagger \circ U \circ T_\Psi\| + \frac{\epsilon\sqrt{B_\Psi}}{\sqrt{A_\Phi}}$$

$$= \|T_\Phi^\dagger\|\|T_\Phi \circ u(\Psi,\Phi) \circ T_\Psi^\dagger - U\|\|T_\Psi\| + \frac{\epsilon\sqrt{B_\Psi}}{\sqrt{A_\Phi}}$$

$$\leq \frac{\epsilon\sqrt{B_\Psi}}{\sqrt{A_\Phi}} + \frac{\epsilon\sqrt{B_\Psi}}{\sqrt{A_\Phi}} = \frac{2\epsilon\sqrt{B_\Psi}}{\sqrt{A_\Phi}},$$

where for the last inequality, similarly as before, we employ that the range of $T_\Phi \circ u(\Psi,\Phi) \circ T_\Psi^\dagger - U$ lies in $\mathcal{M}_\Phi$. This concludes the proof.

## B  Analyzing Operator Learning Architectures

### B.1  Fourier layer in FNOs

We focus here on the Fourier layer of FNOs, i.e.

$$\mathcal{K}v = \mathcal{F}^{-1}(R \odot \mathcal{F})(v), \tag{B.1}$$

where $\mathcal{F}, \mathcal{F}^{-1}$ denote the Fourier transform and its inverse, and where $R$ denotes a low-pass filter. In [17], the authors define the Fourier layer on the space $L^2(\mathbb{T})$ of 2-periodic functions and, with slight abuse of notation, they refer to the mappings $\mathcal{F}\colon L^2(\mathbb{T}) \to \ell^2(\mathbb{Z})$ and $\mathcal{F}^{-1}\colon \ell^2(\mathbb{Z}) \to L^2(\mathbb{T})$,

$$\mathcal{F}w(k) = \langle w, e^{i\pi kx}\rangle, \qquad \mathcal{F}^{-1}(\{W_k\}_{k\in\mathbb{Z}}) = \sum_{k\in\mathbb{Z}} W_k e^{i\pi kx},$$

as the Fourier transform and the inverse Fourier transform. Furthermore, the authors define the discrete version of (B.1) as $F^{-1}(R \odot F)$, where $F, F^{-1}$ denote the discrete Fourier transform (DFT) and its inverse, and where they assume to have access only to point-wise evaluations of the input and output functions. However, *the space of 2-periodic functions is too large to allow for any form of continuous-discrete equivalence (CDE)* when the input $v$ and the output $\mathcal{K}v$ are represented by their point samples. Consequently, here we consider smaller subspaces of $L^2(\mathbb{T})$ which allow for CDEs. More precisely, we are able to show that FNO Fourier layers can be realized as Representation equivalent Operators (crf. Definition 3.4) between bandlimited and periodic functions. Let $K > 0$ and let $\mathcal{P}_K$ be the space of bandlimited 2-periodic functions

$$\mathcal{P}_K = \left\{ w(x) = \sum_{k=-K}^{K} W_k e^{i\pi kx} : \{W_k\}_{k=-K}^{K} \in \mathbb{C}^{2K+1} \right\}.$$

Every function $w \in \mathcal{P}_K$ can be uniquely represented by its Fourier coefficients $\{W_k\}_{k=-K}^{K}$ as well as by its samples $\{w(\frac{k}{2K+1})\}_{k=-K}^{K}$, see [26, Section 5.5.2]. Indeed, the latter ones are the coefficients of $w$ with respect to the orthonormal basis

$$\Psi_K = \left\{ \frac{1}{\sqrt{2(2K+1)}} d\left( \cdot - \frac{2k}{2K+1} \right) \right\}_{k=0}^{2K}, \tag{B.2}$$

where $d$ denotes the Dirichlet kernel of order $K$ and period 2, defined as

$$d(t) = \sum_{k=-K}^{K} e^{i\pi k t}.$$

Furthermore, the DFT $\{\widehat{W}_k\}_{k=-K}^{K}$ of the sample sequence $\{w(\frac{k}{2K+1})\}_{k=-K}^{K}$ is related to the Fourier coefficients of $w$ via the equation

$$\widehat{W}_k = (2K+1)W_k, \quad k = -K, \ldots, K,$$

and we refer to [26, Section 5.5.2] for its proof. Thus, this yields the commutative diagram

$$
\begin{array}{ccc}
\mathcal{P}_K & \xrightarrow{\ \mathcal{F}\ } & \mathbb{C}^{2K+1}, \\
{\scriptstyle T_{\Psi_K}^{\dagger}}\big\downarrow & & \big\uparrow{\scriptstyle \mathrm{Id}} \\
\mathbb{C}^{2K+1} & \xrightarrow{(2K+1)\cdot\mathrm{F}} & \mathbb{C}^{2K+1}
\end{array}
$$

where $T_{\Psi_K}^{\dagger} \colon \mathcal{P}_K \to \mathbb{C}^{2K+1}$ denotes the analysis operator associated to the basis (B.2). Analogously, we can build the commutative diagram

$$
\begin{array}{ccc}
\mathbb{C}^{2K'+1} & \xrightarrow{\ \mathcal{F}^{-1}\ } & \mathcal{P}_{K'} \\
{\scriptstyle \mathrm{Id}}\big\downarrow & & \big\uparrow{\scriptstyle T_{\Psi_{K'}}} \\
\mathbb{C}^{2K'+1} & \xrightarrow{\frac{1}{(2K'+1)}\cdot\mathrm{F}^{-1}} & \mathbb{C}^{2K'+1}
\end{array}
$$

where $T_{\Psi_{K'}} \colon \mathbb{C}^{2K'+1} \to \mathcal{P}_{K'}$ denotes the synthesis operator associated to the basis (B.2) with $K = K'$. Then, $R = \{R_k\}_{k=-K'}^{K'}$, with $K' \le K$, denotes the Fourier coefficients of a 2-periodic function, and the mapping $R \odot \mathcal{F} \colon \mathcal{P}_K \to \mathbb{C}^{2K'+1}$ is defined as

$$(R \odot \mathcal{F}w)(k) = R_k W_k, \quad k = -K', \ldots, K'.$$

Therefore, by definition, $R \odot \mathcal{F}$ yields a continuous-discrete equivalence operation. Overall, we get the commutative diagram

$$
\begin{array}{ccccccc}
\mathcal{P}_K & \xrightarrow{\ \mathcal{F}\ } & \mathbb{C}^{2K+1} & \xrightarrow{\ R\odot\ } & \mathbb{C}^{2K'+1} & \xrightarrow{\ \mathcal{F}^{-1}\ } & \mathcal{P}_{K'} \\
{\scriptstyle T_{\Psi_K}^{\dagger}}\big\downarrow & & \big\uparrow{\scriptstyle \mathrm{Id}} & & \big\downarrow{\scriptstyle \mathrm{Id}} & & \big\uparrow{\scriptstyle T_{\Psi_{K'}}} \\
\mathbb{C}^{2K+1} & \xrightarrow{(2K+1)\cdot\mathrm{F}} & \mathbb{C}^{2K+1} & \xrightarrow{\ R\odot\ } & \mathbb{C}^{2K'+1} & \xrightarrow{\frac{1}{(2K'+1)}\cdot\mathrm{F}^{-1}} & \mathbb{C}^{2K'+1}
\end{array}
$$

which shows that the discretization of the Fourier layer B.1, the blue path in the above commutative diagram, is defined via Equation (3.2). As a consequence, the Fourier layer B.1, regarded as an operator from $\mathcal{P}_K$ into $\mathcal{P}_{K'}$ satisfies the requirements of a Representation equivalent Operator (crf. Definition 3.4). However, as pointed out in 4, the pointwise activation function applied to a bandlimited input will not necessarily respect the bandwidth. In fact, with popular choices of activation functions such as ReLU, $\sigma(f) \notin \mathcal{P}_K$, for any $K \in \mathbb{N}$ (see also **SM C** for numerical illustrations). Thus, the FNO layer

$$\sigma(\mathcal{K}v) = \sigma(\mathcal{F}^{-1}(R \odot \mathcal{F})(v))$$

may not respect the continuous-discrete equivalence and can lead to aliasing errors, a fact already identified in [10]. Hence, FNOs *may not be ReNOs in the sense of Definition 3.4.*

## B.2  Layer in CNO

We start by setting some notation. For every $w > 0$, we denote by $\mathcal{B}_w(\mathbb{R}^2)$ the space of multivariate bandlimited functions

$$\mathcal{B}_w(\mathbb{R}^2) = \{f \in L^2(\mathbb{R}^2) : \operatorname{supp}\hat{f} \subseteq [-w, w]^2\}.$$

The set $\Psi_w = \{\operatorname{sinc}(2wx_1 - m) \cdot \operatorname{sinc}(2wx_2 - n)\}_{m,n\in\mathbb{Z}}$ constitutes an orthonormal basis for $\mathcal{B}_w(\mathbb{R}^2)$.

The convolutional operator appearing in (4.3) takes the form

$$\mathcal{K}_w f(x) = \sum_{m,n=-k}^{k} k_{m,n} f(x - z_{m,n}), \quad x \in \mathbb{R},$$

for some $w > 0$, where $k \in \mathbb{N}$, $k_{m,n} \in \mathbb{C}$ and $z_{m,n} = \left\{\left(\frac{m}{2w}, \frac{n}{2w}\right)\right\}_{m,n\in\mathbb{Z}}$. By definition, $\mathcal{K}_w$ is a well-defined operator from $\mathcal{B}_w(\mathbb{R}^2)$ into itself. Moreover, its discretized version is defined by the mapping

$$\left\{f\left(\frac{m}{2w}, \frac{n}{2w}\right)\right\}_{m,n\in\mathbb{Z}} \to \left\{\mathcal{K}_w f\left(\frac{m}{2w}, \frac{n}{2w}\right)\right\}_{m,n\in\mathbb{Z}} = \left\{\sum_{m',n'=-k}^{k} k_{m',n'} f(z_{m,n} - z_{m',n'})\right\}_{m,n\in\mathbb{Z}},$$

and thus results in the commutative diagram

$$
\begin{array}{ccc}
\mathcal{B}_w & \xrightarrow{\ \mathcal{K}_w\ } & \mathcal{B}_w \\
{\scriptstyle T_{\Psi_w}}\uparrow & & \downarrow{\scriptstyle T^*_{\Psi_w}} \\
\ell^2(\mathbb{Z}^2) & \longrightarrow & \ell^2(\mathbb{Z}^2)
\end{array}
$$

Equivalently, the discretized verion of $\mathcal{K}_w$ is defined via (3.2), which was to be shown. In order to define the activation layer $\Sigma_l$, we first assume that the activation function $\sigma \colon \mathbb{R}^2 \to \mathbb{R}^2$ is such that for every $f \in \mathcal{B}_w(\mathbb{R}^2)$

$$\sigma(f) \in \mathcal{B}_{\overline{w}}(\mathbb{R}^2), \tag{B.3}$$

for some $\overline{w} > w$. In fact, in [23] the authors assume that the pointwise activation can be approximated by an operator between bandlimited spaces and consequently (B.3) is satisfied up to negligible frequencies. Thus, the activation layer $\Sigma_{w,\overline{w}} \colon \mathcal{B}_w(\mathbb{R}^2) \to \mathcal{B}_w(\mathbb{R}^2)$ in (4.3) is defined by the composition

$$\Sigma_{w,\overline{w}} = P_{\mathcal{B}_w(\mathbb{R}^2)} \circ \sigma \circ P_{\mathcal{B}_{\overline{w}}(\mathbb{R}^2)}, \tag{B.4}$$

where $P_{\mathcal{B}_w(\mathbb{R}^2)} \colon \mathcal{B}_{\overline{w}}(\mathbb{R}^2) \to \mathcal{B}_w(\mathbb{R}^2)$ denotes the orthogonal projection onto $\mathcal{B}_w(\mathbb{R}^2)$ and $P_{\mathcal{B}_{\overline{w}}(\mathbb{R}^2)} \colon \mathcal{B}_w(\mathbb{R}^2) \to \mathcal{B}_{\overline{w}}(\mathbb{R}^2)$ denotes the natural embedding of $\mathcal{B}_w(\mathbb{R}^2)$ into $\mathcal{B}_{\overline{w}}(\mathbb{R}^2)$. The discretized version of each mapping in (B.4) is defined in order to guarantee a continuous-discrete equivalence between the continuous and discrete levels. More precisely, $P_{\mathcal{B}_w(\mathbb{R}^2)}$ and $P_{\mathcal{B}_{\overline{w}}(\mathbb{R}^2)}$ are discretized via (3.2) as

$$\mathcal{D}_{\overline{w},w} = T^*_{\Psi_w} \circ P_{\mathcal{B}_w(\mathbb{R}^2)} \circ T_{\Psi_{\overline{w}}}, \qquad \mathcal{U}_{w,\overline{w}} = T^*_{\Psi_{\overline{w}}} \circ P_{\mathcal{B}_{\overline{w}}(\mathbb{R}^2)} \circ T_{\Psi_w},$$

which are respectively called downsampling and upsampling. Consequently, the discretized version of the activation layer is given by the composition

$$\mathcal{D}_{\overline{w},w} \circ \sigma \circ \mathcal{U}_{w,\overline{w}},$$

which yields the commutative diagram

$$
\begin{array}{ccccccc}
\mathcal{B}_w & \xleftarrow{P_{\mathcal{B}_{\overline{w}}(\mathbb{R}^2)}} & \mathcal{B}_{\overline{w}} & \xrightarrow{\ \sigma\ } & \mathcal{B}_{\overline{w}} & \xrightarrow{P_{\mathcal{B}_w(\mathbb{R}^2)}} & \mathcal{B}_w \\
{\scriptstyle T_{\Psi_w}}\uparrow & & \downarrow{\scriptstyle T^*_{\Psi_{\overline{w}}}} & & {\scriptstyle T_{\Psi_{\overline{w}}}}\uparrow & & \downarrow{\scriptstyle T^*_{\Psi_w}} \\
\ell^2(\mathbb{Z}^2) & \xrightarrow{\mathcal{U}_{w,\overline{w}}} & \ell^2(\mathbb{Z}^2) & \xrightarrow{\ \sigma\ } & \ell^2(\mathbb{Z}^2) & \xrightarrow{\mathcal{D}_{\overline{w},w}} & \ell^2(\mathbb{Z}^2)
\end{array}
$$

which we wanted to show. Finally, the activation layer might be followed by an additional projective operator, i.e., by a downsampling or an upsampling. Thus, this exact correspondence between its constituent continuous and discrete operators establishes CNO as an example of Representation equivalent neural operators or ReNOs in the sense of Definiton 3.4. It is worth observing that the above proofs can be readily adapted to bandlimited periodic functions, i.e. periodic functions with a finite number of non-zero Fourier coefficientswith the Dirichlet kernel as a counterpart of the sinc function, see [26, Section 5.5.2] for further details.

## B.3 DeepONets

Following [20] and for simplicity of exposition, we consider *sensors*, located on a uniform grid on $[-1, 1]$ with grid size $1/(2N+1)$. We assume that the input function $f \in \mathcal{P}_N$, where $\mathcal{P}_N$ is defined as in **SM** B.1 with orthonormal basis $\Psi_N$ chosen according to (B.2). Denote the corresponding synthesis and analysis operators as $T_{\Psi_N}, T^*_{\Psi_N}$, respectively. Let $\tau_k : \mathbb{R} \to \mathbb{R}$, for $1 \leq k \leq K$ be neural networks that form the so-called *trunk nets* in a DeepONet [20] and denote $\mathcal{Q}_K = \text{span}\{\tau_k : k = 1, \ldots, K\}$. Clearly $\mathcal{T}_K = \{\tau_k\}_{k=1}^K$ constitutes a *frame* for $\mathcal{Q}_K$ and we can denote the corresponding synthesis and analysis operators by $T_{\mathcal{T}_K}, T^*_{\mathcal{T}_K}$. Then a DeepONet [20] is given by the composition $T_{\mathcal{T}_K} \circ \mathcal{N} \circ T^*_{\Psi_N}$, with $\mathcal{N} : \mathbb{R}^{2N+1} \to \mathbb{R}^K$, being a neural network of the form (B.5) that is termed as the *branch net* of the DeepONet. Written in this manner, DeepONet satisfies Definition 3.4 as it corresponds to the following commutative diagram,

$$
\begin{array}{ccccccc}
\mathcal{P}_N & \xrightarrow{T^*_{\Psi_N}} & \mathbb{R}^{2N+1} & \xrightarrow{\mathcal{N}} & \mathbb{R}^K & \xrightarrow{T_{\mathcal{T}_K}} & \mathcal{Q}_K \\
{\scriptstyle T_{\Psi_N}} \big\updownarrow {\scriptstyle T^*_{\Psi_N}} & & \big\uparrow {\scriptstyle \text{Id}} & & \big\uparrow {\scriptstyle \text{Id}} & {\scriptstyle T_{\mathcal{T}_K}} \big\updownarrow {\scriptstyle T^*_{\mathcal{T}_K}} \\
\mathbb{R}^{2N+1} & \xrightarrow{\text{Id}} & \mathbb{R}^{2N+1} & \xrightarrow{\mathcal{N}} & \mathbb{R}^K & \xrightarrow{\text{Id}} & \mathbb{R}^K
\end{array}
$$

However, it is essential to emphasize that the choice of the underlying function spaces is essential in regard to the Definition 3.4 of Representation equivalent Neural Operators. For instance, if the sensors are *randomly* distributed rather than located on a uniform grid, then it can induce aliasing errors for DeepONets (see also [16] for a discussion on this issue). Similarly, DeepONets are only ReNOs with respect to the function space $\mathcal{Q}_K$ as the target space. Changing the target function space to another space, say for instance $\mathcal{P}_{N'}$, for some $N'$, will lead to aliasing errors as the trunk nets do not necessarily form a frame for the space of bandlimited 2-periodic functions.

## B.4 Spectral Neural Operators (SNO)

Introduced in [10], this architecture is defined as follows. Let $K > 0$ and denote by

$$
\mathcal{P}_K = \left\{ g(x) = \sum_{k=-K}^{K} c_k e^{i\pi k x} : \{c_k\}_{k=-K}^K \in \mathbb{C}^{2K+1} \right\},
$$

the space of 2-periodic signals bandlimited to $\pi K$. Clearly, $\Psi_K = \{e^{i\pi k\cdot}\}_{k=-K}^K$ constitutes an orthonormal basis for $\mathcal{P}_K$, and the corresponding synthesis operator $T_{\Psi_K} : \mathbb{C}^{2K+1} \to \mathcal{P}_K$ and analysis operator $T^*_{\Psi_K} : \mathcal{P}_K \to \mathbb{C}^{2K+1}$ are given by

$$
T_{\Psi_K}(\{c_k\}_{k=-K}^K) = \sum_{k=-K}^{K} c_k e^{i\pi k\cdot}, \quad T^*_{\Psi_K} f = \{\langle f, e^{i\pi k\cdot}\rangle\}_{k=-K}^K.
$$

A spectral neural operator is defined as the compositional mapping $T_{\Psi_{K'}} \circ \mathcal{N} \circ T^*_{\Psi_K}$, where $\mathcal{N} : \mathbb{C}^{2K+1} \to \mathbb{C}^{2K'+1}$ is an ordinary feedforward neural network with activation function $\sigma$,

$$
\mathcal{N}(x) = W^{(L+1)} \sigma(W^{(L)} \cdots \sigma(W^{(2)} \sigma(W^{(1)} x - b^{(1)}) - b^{(2)}) \cdots - b^{(L)}) - b^{(L+1)} \tag{B.5}
$$

for some weights $W^{(\ell)}$ and biases $b^{(\ell)}$, $\ell = 1, \ldots, L+1$. It is straightforward to see that this architecture corresponds to the commutative diagram,

$$
\begin{array}{ccccccc}
\mathcal{P}_K & \xrightarrow{T^*_{\Psi_K}} & \mathbb{C}^{2K+1} & \xrightarrow{\mathcal{N}} & \mathbb{C}^{2K'+1} & \xrightarrow{T_{\Psi_{K'}}} & \mathcal{P}_{K'} \\
{\scriptstyle T_{\Psi_K}} \big\updownarrow {\scriptstyle T^*_{\Psi_K}} & & \big\uparrow {\scriptstyle \text{Id}} & & \big\uparrow {\scriptstyle \text{Id}} & {\scriptstyle T_{\Psi_{K'}}} \big\updownarrow {\scriptstyle T^*_{\Psi_{K'}}} \\
\mathbb{C}^{2K+1} & \xrightarrow{\text{Id}} & \mathbb{C}^{2K+1} & \xrightarrow{\mathcal{N}} & \mathbb{C}^{2K'+1} & \xrightarrow{\text{Id}} & \mathbb{C}^{2K'+1}
\end{array}
$$

A discretized version of spectral neural operators simply corresponds to an ordinary feedforward neural network mapping Fourier coefficients to Fourier coefficients. We conclude that for SNOs to be ReNOs with respect to the function spaces $\mathcal{P}_K, \mathcal{P}'_K$, we have to enforce that for more general choices of frame sequences, equation (3.3) is satisfied. Moreover, the architecture of SNOs can be generalized with respect to any frames in finite-dimensional inner-product spaces.

## C Empirical Analysis

### C.1 Illustration of the effect of the activation function

We consider the same setting as for the FNO in Section 4 and **SM** B.1, i.e. considering $f \in \mathcal{P}_K, K = 20$ to be both periodic and bandlimited. On the upmost plot of Figure 3, we observe the values of $f$, as well as relu($f$) and gelu($f$). In this case, pointwise samples on the grid of size $2K + 1$ are enough to characterize $f$, as its Fourier coefficients are zero above the Nyquist frequency $K$. However, as we can clearly observe on the lower plot, this is no longer the case for functions relu($f$) and gelu($f$), and as a consequence the continuous functions are no longer represented uniquely on the grid, and aliasing errors occur.

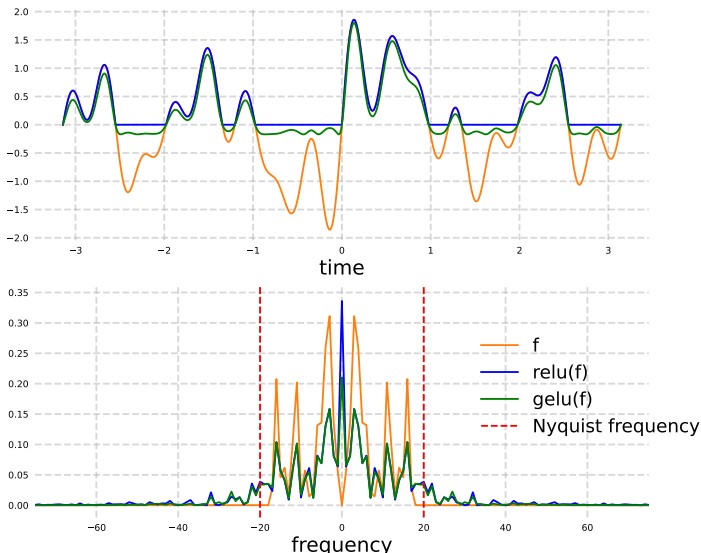

Figure 3: The activation function increases the bandwidth of the input function beyond the Nyquist frequency, causing aliasing errors.

### C.2 Additional Details for Experimental Analysis Section 5.1

On the continuous level, SNO maps from $H$ into $H$. Thus, at the discrete level, $u(\Psi_M, \Psi_M)$ corresponds to the following: it takes in $2M + 1$ point samples, synthesizes these to a function in $H$ which is then sampled on the training grid. Then, $u(\Psi_K, \Psi_K)$ is applied to this input vector of length $2K + 1$. Finally, the output vector is synthesized to a function in $H$ and then sampled on the evaluation grid. *PCANetJitter* models the fact that there may be slightly different data at each resolution. This is done by performing a PCA at each resolution, randomly removing one data sample beforehand. This is due to the fact that even a small change of ordering in the eigenfunctions can introduce large errors.

### C.3 Minimizing representation equivalence errors

In this section, instead of directly tackling aliasing errors from a theoretical perspective, we select an architecture that may have aliasing, i.e. FNO, and we try to minimize the representation equivalence error during training. To this extent, we repeat the same experiment as with FNO Section 5.1, aiming now to minimize both the regression loss and the representation equivalence error simultaneously, introducing a multiplier $\lambda$ on the latter term. For each batch we have the total loss:

$$l(u) = \|(u(\Psi_K, \Psi_K) - u_K^*\|_1 + \lambda \cdot \tau(u(\Psi_K, \Psi_K), u(\Psi_M, \Psi_M)), \qquad (C.1)$$

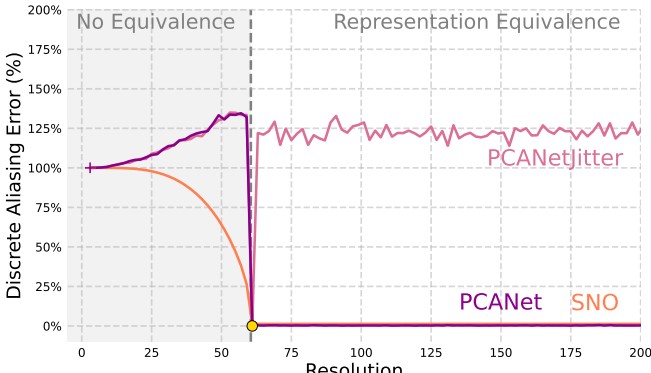

Figure 4: Representation equivalence analysis is conducted by training SNO, PCANet, and PCANet with jittering on a fixed resolution and examining their performance when input resolution is varied. The "Representation Equivalence" zone, located on the right-hand side, denotes the region where discrete representations have an associated frame, while the left-hand side represents the area where this is no longer the case.

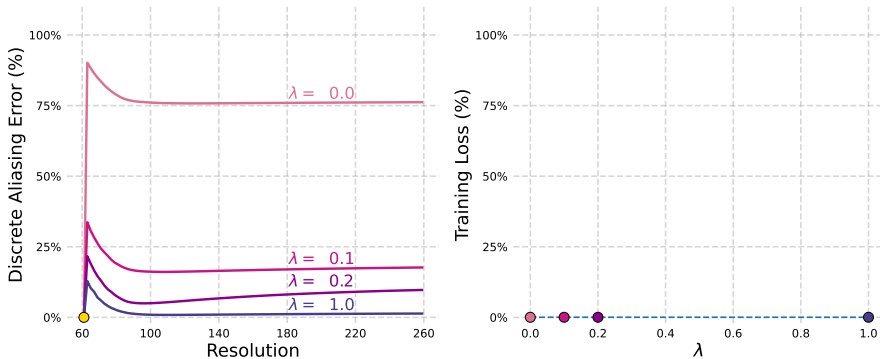

Figure 5: Minimizing the representation equivalence error. We compare FNOs trained by minimizing different weighted combinations of the supervised and discrete aliasing error. We observe reduced aliasing for all resolutions when $\lambda$ increases, without much change in training loss.

where $M \sim \text{Unif}\{K, \dots, 2K\}$. The outcomes are illustrated in Figure 5. These results showcase that as $\lambda$ increases, the aliasing error is indeed minimized for $M \in [K, 2K]$. Remarkably, for $\lambda = 1$, the error is almost negligible, extending beyond the range observed during training to $M > 2K$. This could perhaps be due to the fact that the representation equivalence error and operator aliasing are linked, and that by minimizing the representation equivalence error, we are also minimizing aliasing errors as well. Additionally, the training error depicted in Fig. 5 (right) remains small even with increasing values of $\lambda$, indicating no discernible trade-off between approximation error and aliasing error for FNO.

# D   An Introduction to Frame Theory

Let $\mathcal{H}$ be a separable Hilbert space with inner product $\langle \cdot, \cdot \rangle$ and norm $\| \cdot \|$. A countable sequence of vectors $\{f_i\}_{i \in I}$ in $\mathcal{H}$ is a *frame* for $\mathcal{H}$ if there exist constants $A, B > 0$ such that for all $f \in \mathcal{H}$

$$A\|f\|^2 \leq \sum_{i \in I} |\langle f, f_i \rangle|^2 \leq B\|f\|^2.$$

We say that $\{f_i\}_{i \in I}$ is a *tight frame* if $A = B$ and, in particular, a *Parseval frame* if $A = B = 1$. Clearly, by the Parseval identity, an orthonormal basis for $\mathcal{H}$ is a Parseval frame. The lower inequality implies that

$$\langle f, f_i \rangle = 0, \ \forall i \in I \implies f = 0,$$

which is equivalent to

$$\overline{\text{span}}\{f_i : i \in I\} = \mathcal{H}.$$

On the other hand, the upper inequality implies that the operator

$$T \colon \ell^2(I) \to \mathcal{H}, \quad T(\{c_i\}_{i \in I}) = \sum_{i \in I} c_i f_i,$$

is bounded with $\|T\| \le \sqrt{B}$ [7, Theorem 3.1.3], and we call $T$ the *synthesis operator*. Its adjoint is given by

$$T^* \colon \mathcal{H} \to \ell^2(I), \quad T^* f = (\langle f, f_i \rangle)_{i \in I},$$

[7, Lemma 3.1.1] and is called the *analysis operator*. By composing $T$ and $T^*$, we obtain the *frame operator*

$$S \colon \mathcal{H} \to \mathcal{H}, \quad S f = T T^* f = \sum_{i \in I} \langle f, f_i \rangle f_i,$$

which is a bounded, invertible, self-adjoint and positive operator [7, Lemma 5.1.6]. We note that the frame operator is invertible since it is bounded, being a composition of two bounded operators, and the frame property implies that $\| \text{Id} - B^{-1} S \| < 1$, where $\text{Id}$ denotes the identity operator. Furthermore, the pseudo-inverse of the synthesis operator is given by

$$T^\dagger \colon \mathcal{H} \to \ell^2(I), \quad T^\dagger f = (\langle f, S^{-1} f_i \rangle)_{i \in I},$$

[7, Theorem 5.3.7] and $\|T^\dagger\| \le 1/\sqrt{A}$ [7, Proposition 5.3.8]. The composition $T T^\dagger$ gives the identity operator on $\mathcal{H}$, and consequently every element in $\mathcal{H}$ can be reconstructed via the reconstruction formula

$$f = T T^\dagger f = \sum_{i \in I} \langle f, S^{-1} f_i \rangle f_i = \sum_{i \in I} \langle f, f_i \rangle S^{-1} f_i, \tag{D.1}$$

where the series converge unconditionally. Formula (D.1) is known as the *frame decomposition theorem* [7, Theorem 5.1.7]. In particular, if $\{f_i\}_{i \in I}$ is a tight frame, then $S = A \, \text{Id}$ and formula (D.1) simply reads

$$f = \frac{1}{A} \sum_{i \in I} \langle f, f_i \rangle f_i.$$

On the other hand, the composition $T^\dagger T$ gives the orthogonal projection of $\ell^2(I)$ onto $\text{Ran} \, T^\dagger$ [7, Lemma 2.5.2].

In what follows, we consider sequences which are not complete in $\mathcal{H}$, and consequently are not frames for $\mathcal{H}$, but they are frames for their closed linear span.

**Definition D.1.** *Let* $\{v_i\}_{i \in I}$ *be a countable sequence of vectors in* $\mathcal{H}$. *We say that* $\{v_i\}_{i \in I}$ *is a* frame sequence *if it is a frame for* $\overline{\text{span}}\{v_i : i \in I\}$.

A frame sequence $\{v_i\}_{i \in I}$ in $\mathcal{H}$ with synthesis operator $T \colon \ell^2(I) \to \overline{\text{span}}\{v_i : i \in I\}$ is a frame for $\mathcal{H}$ if and only $T^*$ is injective, whilst in general $T^*$ is not surjective and consequently $T$ is not injective. We denote $\mathcal{V} = \overline{\text{span}}\{v_i : i \in I\}$. Then, the orthogonal projection of $\mathcal{H}$ onto $\mathcal{V}$ is given by

$$\mathcal{P}_\mathcal{V} f = T T^\dagger = \sum_{i \in I} \langle f, S^{-1} v_i \rangle v_i,$$

where $S \colon \mathcal{V} \to \mathcal{V}$ denotes the frame operator. Hence, reconstruction formula (D.1) holds if and only if $f \in \mathcal{V}$.

## D.1  Condition $u\colon \operatorname{Ran} T_\Psi^\dagger \to \operatorname{Ran} T_\Phi^\dagger$

Once we choose how to discretize the functions in the input and output spaces, which amounts to choosing a frame pair $(\Psi \subseteq \mathcal{H}, \Phi \subseteq \mathcal{K})$, we want to define a mapping $u\colon \ell^2(I) \to \ell^2(K)$ which handles such discrete representations. Notice that, by the frame decomposition theorem (D.1), every function in $\mathcal{H}$ is uniquely determined by a sequence in $\operatorname{Ran} T_\Psi^\dagger$, and analogously every function in $\mathcal{K}$ is uniquely determined by a sequence in $\operatorname{Ran} T_\Phi^\dagger$. It is therefore sufficient to define $u$ as a mapping from $\operatorname{Ran} T_\Psi^\dagger$ into $\operatorname{Ran} T_\Phi^\dagger$. By enforcing this condition, we ensure that when two different discretizations both yield zero aliasing, indeed (3.3) holds, and thus the representation equivalence error is zero.

## E  Depiction of a ReNO architecture

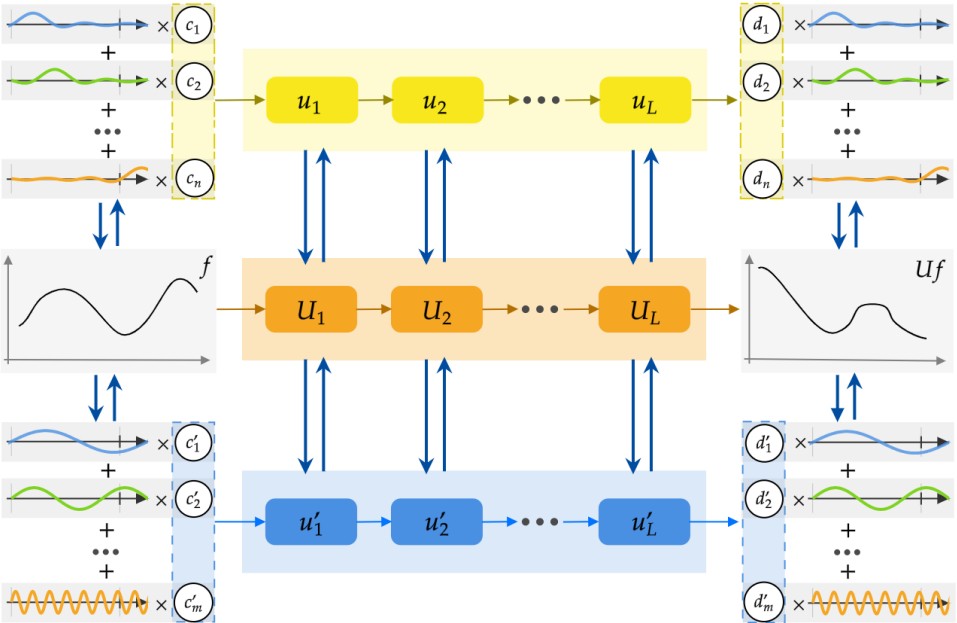

Figure 6: Sketch of the ReNO framework. The learned operator $U_L \circ \ldots U_2 \circ U_1$ on the continuous level must be realized by discrete operations. Any discrete representation $u_L \circ \ldots u_2 \circ u_1$ corresponds to the continuous level by a stable 1-to-1 correspondence (blue arrows) between $U_i$ and $u_i$ for each layer $i$. In this way, any two discretizations $u$ and $u'$ are also linked.