# OpenReview forum: "Representation Equivalent Neural Operators: a Framework for Alias-free Operator Learning"
_NeurIPS.cc/2023/Conference — NeurIPS 2023 poster_

### Official Review · Reviewer_ydDu · 2023-07-06

**Soundness:** 3 good
**Presentation:** 2 fair
**Contribution:** 3 good
**Rating:** 7
**Confidence:** 3

**Summary:**

The paper proposes a new paradigm to look at operator learning via the lens of frame theory termed as Representation Equivalent Neural Operator (ReNO). It talks about an important missing piece in the literature, the one regarding the balance of continuous nature of operators and the discrete nature of data these are trained on. Aliasing error is used as the quantifier for determining the equivalence.

**Strengths:**

The paper is very well motivated and suitably placed in the literature.

The use of frames as a tool for representation is presented in an interesting way.

The paper address a common issue in a very mathematically rigorous manner.

The experimental evidence is convincing.




**Weaknesses:**

The overall presentation of the paper can be improved significantly.

Few pointers: Figures appearing the paper have no caption, Figure 1,2 and 3 have been referred within the text, others not. In a paper like these, visual guides are very much helpful, otherwise it is easy to not follow. Additionally there is over use of inline equations which are hard to parse. If space is the issue, some details can be moved out to the appendix, for example section 2 can be compressed, although it reads well, it is somewhat standard.

Notations, need to made clear around when introduced. For example: that $\mathfrak{g}$ is dicretization of $\mathcal{G}$ is mentioned around line 211 and introduced much earlier. This makes it harder to read through.

Experimental section should have more details, the current form is not at all clear in the main paper. I see lot of blank space in that page anyway.

**Questions:**

Of all the three architectures considered in the paper, none of them perform on resolutions lower than the one used for training. Can the authors discuss this a bit?

While the paper is written in general for frames, can the authors comment on some pros/cons of specific choice for example wavelet frames? That would be very interesting.

**Limitations:**

I don't see any direct potential negative social impact.

Authors have mentioned technical limitations which is very much appreciated.

---

> ### Author Rebuttal · Authors · 2023-08-07
>
> We start by thanking the reviewer for their appreciation of the merits of our paper and their welcome suggestions to improve it. We address their detailed concerns below.
>
> 1.  In a CRV, if accepted, we will ensure that all figures have captions and are referenced within the main text.
>
> 2. The reviewer's point about *visual guides being important* is excellent and we follow this suggestion. As a visual guide, in the CRV, we will add a cartoon depiction of the ReNO framework  that we have created in Figure 1 in the uploaded pdf for this rebuttal.
>
> 3. We will also reduce the number of inline equations and move parts of Section 2 to the appendix if necessary.
>
> 4. The definition of $\mathfrak{g}$ is stated in line 191, just shortly after the mentioned instance in line 211, and its definition is also a numbered equation that we refer to in line 211.
>
> 5. Regarding the reviewer's concern about lack of experimental details, we kindly refer to our answer in the general response to all the reviewers. With this, we hope to clarify the questions about the experiments and we would add such explanation in a CRV.
>
> 6. When testing below training resolution in our experiment, indeed, any architecture will yield poor results. The input and output space in these experiments are spaces of bandlimited periodic function of bandwidth $K$. If we then test with resolution $M<K$, we are possibly undersampling the functions: if a function has bandwidth $K$, we will need at least $2K+1$ coefficients to represent it. At resolution $M$, only $2M+1<2K+1$ point samples of the function are taken. This automatically introduces errors as it means an attempt to represent a $2K+1$-dimensional vector space by only $2M+1<2K+1$ elements. We hope that this clarifies the reviewer's question and we will make it point to add this explanation to a CRV, if accepted.
>
> 7. Wavelet systems can be constructed to yield frames for $L^2$ and in the context of ReNOs, this setup would enjoy the  property that applications of nonlinear activation functions naturally remain in the space spanned by the frame. In many applications, however, the measurements are given as point evaluations rather than wavelet coefficients, so that some transformation would have to be performed initially. Also, an architecture based on a multiresolution analysis such as wavelets, curvelets, shearlets, etc. is in general more technical (e.g. separate channel for each scale). These difficulties can of course be overcome so that we believe wavelets to be an interesting family of frames for the ReNO framework that we plan to study in future work on designing ReNO architectures.
>
> We sincerely hope to have addressed your concerns, particularly about the presentation of the paper, satisfactorily and would kindly request the reviewer to update their assessment accordingly.

---

> > ### Comment · Reviewer_ydDu · 2023-08-18
> >
> > Thank you authors for your response. Please try to add the discussion regarding resolution, point 6 above to the updated version of the paper. My concerns have been address and I am increasing my score.

---

> > > ### Author Response · Authors · 2023-08-18
> > > **Thanking the Reviewer**
> > >
> > > We thank the reviewer for your positive feedback and for increasing our score. We will certainly incorporate our reply to pt 6 in a CRV, if accepted. It is a very valid point that needs to be made.

---

### Official Review · Reviewer_2w9i · 2023-07-06

**Soundness:** 2 fair
**Presentation:** 2 fair
**Contribution:** 2 fair
**Rating:** 6
**Confidence:** 2

**Summary:**

Many recent studies have emerged in the field of operator learning. However, many models attempt to learn operators using the discretized values of functions rather than sending functions as operators to other functions. In this paper, the authors interpret the relationship between infinite-dimensional functions and their discretized values using the framework of frame theory. The authors analyze various operator learning models to assess whether they effectively capture this relationship in an equivalent manner, and they propose a framework called ReNO for this purpose.

**Strengths:**

Investigating whether models that engage in operator learning, a mapping process between infinite-dimensional function spaces, effectively handle and address this infinite dimensionality is a pivotal and highly relevant topic within the field. The endeavor to analyze and comprehend this specific aspect of operator learning appears to be a novel and significant contribution. Leveraging the definitions associated with frame theory, as established in Hilbert spaces, and extending their application to operators, enables a profound exploration of the extent to which various operator models align with the ReNO framework. This comprehensive analysis sheds light on the capacity of these models to capture and interpret the intricate relationship between infinite-dimensional functions and their discretized counterparts. Such insights provide valuable guidance for the advancement of operator learning methodologies and contribute to the broader understanding of this field.

**Weaknesses:**

The explanations of frame theory discussed in Sections 2 and 3, as well as the summaries of applying frame theory to operators, are written in a manner that is not easily comprehensible. Many readers, including myself, who may lack a strong background in signal processing and functional analysis, find these sections challenging to understand (See Questions). It would be beneficial to provide clearer explanations and examples regarding aspects such as the invertibility of the frame operator S and the well-definedness of the pseudo-inverse, clarifying their significance.

Furthermore, in Section 4, which aims to verify whether the defined operator learning models conform to ReNO, it is intuitive to consider that CNN and FNO may not fall under the purview of ReNO, as they simply discretize functions into image-like forms. On the other hand, SNO, which predefines a basis and solely learns the mapping between the coefficients, appears to naturally align with ReNO. It would be intriguing to explore additional insights derived from these theories based on this intuition. Additionally, there is a well-known operator learning model called DeepONet, which could be described in more detail. While it is briefly mentioned on line 271 that DeepONet also belongs to SNO, further elaboration is necessary to understand how and what differentiates it from other models. Is there a specific reason for focusing solely on CNN, FNO, and SNO among the numerous operator learning models?

Moreover, in Section 5, the experiments conducted, particularly those related to super-resolution discussed in the FNO paper, and the results presented in Figure 4, raise questions regarding whether they truly represent the best experiments to validate the aforementioned theories. Specifically, what are the specific definitions of the frames $\Phi$ and $\Psi$ used in the experiment where the resolution is altered in the predictions (as explained in Section 4 of the FNO paper)? It is unclear whether these definitions are explicitly provided or not. Are there alternative approaches, apart from altering resolution, to demonstrate the equivalence of arbitrary frames $\Phi$ and $\Psi"? It would be valuable to explore additional experimental setups that further support the underlying theories, as relying solely on the resolution experiment appears inadequate.

**Questions:**

* Is there a specific reason why not all the various models mentioned in line 30 were examined under ReNO? Why was the analysis limited to only CNN, FNO, and SNO?

* What are the definitions of $f(nT)$ and $f(n/2\Omega)$ mentioned in line 70? Additionally, why do the sinc functions in line 75 form an orthonormal basis? Understanding these concepts seems crucial to grasp the difference between Definition 2.1's $f$ and $P_B f$.

* I'm curious to understand why the adjoint $T^*$ in line 93 is well-defined uniquely for all $f_i$. Furthermore, what makes the frame operator $S$ invertible and well-defined?

* In line 102, when referring to "point samples of the underlying function $f$," which specific $f_i$ does this pertain to? (Similarly, for line 221, when does $T^\dagger_\Psi$ precisely become a discretization representation given $\Psi$?)

* I'm interested in understanding how Definitions 2.1 and 2.2 naturally lead to the discussion of aliasing error in Definition 3.1. Explaining the differences and similarities between these concepts would be helpful.

* The diagrams between lines 156 and 157, the diagram mentioned below line 265, and the captions and explanations for Figures 1, 2, and 3 are significantly lacking. They need to be added with essential details. What are the differences between the three figures, and how are they interconnected?

* What is the relevance of Equation 4.3 in the analysis of SNO, considering the diagram? While A and B were assumed to be zero in FNO, does this equation account for the bias term b?

* DeepONet is known as a successful operator learning model mentioned in line 271. What does it mean that DeepONet is also equivalent to SNO? Further elaboration is needed.

* In Section 5, it would be beneficial to summarize the experimental setups (Line 506 F.2) , even briefly in the main text, rather than relegating them to the appendix. It would be helpful to explain what type of regression was performed, whether it involved the solution operator of PDE or not, or if it was a different type of experiment.

**Limitations:**

* The explanation of frame theory is challenging to understand, and the extension of frame theory to operators is equally difficult to grasp. There are intricate aspects in the theoretical part that make it hard to follow, such as meticulous definitions of notations, the reasons for orthogonality, the existence of inverses, whether $T^*$ is necessary for operator extension, and the well-definedness of $T^\dagger$. It would be beneficial to summarize the essential concepts needed to extend these aspects to operators and provide a slightly more accessible explanation. Ultimately, it would be helpful to clearly articulate why frame theory is crucial for explaining operators, rather than presenting it as a separate theory.

* There seems to be a lack of experimental evidence supporting the theoretical claims. In Section 5, which focuses on experiments and analysis related to resolution, additional experiments that better illustrate the theory could enhance the persuasiveness of the need for this theory.

---

> ### Author Rebuttal · Authors · 2023-08-07
>
> We start by thanking the reviewer for their welcome suggestions to improve our paper. We address their detailed concerns below.
>
> 1. We apologize for the lack of clarity in the explanation of frame theory. To address this shortcoming and following the excellent suggestion of the reviewer, we suggest to add a new section  *Frame Theory* in the SM dedicated to the mathematical background on frame theory, including examples of frames for some classical Hilbert spaces and explanations of properties such as the well-definedness and invertibility of the frame operator and the well-definedness of the pseudoinverse. We note that the frame operator $S$ is invertible because it is bounded and one can show that the frame property implies $\\|Id-S\\|<1$.
> 2. To further improve on our exposition, we will add a cartoon depiction of the ReNO framework as a visual guide in the CRV, if accepted. We have created such a scheme in Fig 1 in the uploaded pdf for this rebuttal.
> 3. We agree that DeepONet is an important operator learning architecture. It was already discussed and highlighted in Appendix E, lines 480--496.
> 4. Regarding the reviewer’s concern about lack of experimental details, we kindly refer to our answer in the general response to all reviewers. With this we hope to clarify the questions about the experiments and we would add such an explanation in Section 5 of a
> CRV. As for experiments with other frames, where coefficients do not correspond to point samples: we thank the reviewer for this useful feedback. Preliminary experiments have confirmed that similar results hold in other settings such as wavelet frames and we would add such experiments in the appendix of a CRV, if accepted.
> 5. Fig 2 in the uploaded pdf for this rebuttal reports on an additional experiment we have conducted. It highlights that aliasing error can in fact be eliminated without compromising the approximation power of the network. This further supports the necessity of constructing architectures that respect the representation equivalence. Moreover, we have also explored PCANet of Bhattacharya et al and CNO of Raonic et al arXiv:2302.01178 as additional ReNO architectures (please see General Response to all reviewers and Figs 3 and 4 of attached pdf)
> 6. While our experiments do not involve the solution of a PDE, CNO, which is a ReNO, is demonstrated to be the SOTA architecture for a wide range of operators for benchmark PDEs.
> 7. The expressions $f(nT)$ and $f(n/2\Omega)$ denote evaluation of the function $f$ at the specified points. For example, if $\Omega=1$ and $T=1/2$, then the expressions become $f(nT)=f(n/2\Omega)=f(n/2)$ for $n\in \mathbb{Z},$ resulting in the sequence $(\dots,f(-5/2),f(-3/2),f(-1/2),f(1/2),f(3/2),\dots)$.
> 8. Intuition behind why the $\mbox{sinc}$ functions form a basis for bandlimited functions: Bandlimited functions have Fourier transforms in $L^2([-\Omega,\Omega])$. Consequently, these Fourier transforms admit a Fourier series expansion. The appearing complex exponentials in the Fourier series are related to the translated $\mbox{sinc}$ functions via the Fourier transform.  For a rigorous proof, we refer to the WSK sampling theorem Unser Proc. IEEE, 2000.
> 9. The adjoint $T^*$ is defined as the bounded linear operator such that
> $\langle h,T(\\{c_i\\}\_{i \in I}) \rangle_{\mathcal{H}} = \langle T^* h,  \\{c_i\\}\_{i \in I} \rangle_{\ell^2(I)},$
> for all $h \in \mathcal{H}$ and all $\\{c_i\\}_{i \in I} \in \ell^2(I)$. The fact that $T^*$ takes the form in line 93 follows from Riesz' representation theorem (see Christensen 2008, Lemma 3.1.1).
> 10. In line 102, the general frame coefficients of a function $f$ for a frame $\\{f_i\\}_{i \in I}$ refer to the expressions $\langle f, f_i \rangle$. In the specific case of $\\{f_i\\}\_{i \in I}$ the orthonormal basis of sinc functions, the frame coefficients become point evaluations of $f$, i.e. $\langle f, f_i \rangle = f(i/2\Omega)$ and $I=\mathbb{Z}$ (cf. also Eqn. (2.2)).
> 11. In line 221, since $\Psi$ is a frame for Dom $U$ (cf. Eqn (3.2)) and $u_i \in$ Dom $U$, we obtain that $T_\Psi^\dagger u_i$ is a discrete representation of $u_i$. This means that $u_i$ is obtained by applying $T_\Psi$ to $T_\Psi^\dagger u_i$, i.e. $u_i = T_\Psi T_\Psi^\dagger u_i$ (cf. Eqn (2.4)).
> 12. Definition 2.2 describes the continuous-discrete equivalence between functions in a separable Hilbert space $\mathcal{H}$ and their discrete representation given by frame coefficients w.r.t. a frame sequence $\Psi$ for $\mathcal{H}$. The aliasing error function $\epsilon(f)$ defined therein quantifies the loss of information in representing $f$ by its frame coefficients and can be equivalently expressed as $\epsilon(f)=\operatorname{Id}(f)-T_\Psi\circ \operatorname{id}\circ T_{\Psi}^\dagger(f)$ (cfr. formula in Def. 2.2), where $\operatorname{Id}$ and $ \operatorname{id}$ denote resp. the identity operators of $\mathcal{H}$ and $\ell^2(I)$, with $I$ the index set of $\Psi$. It therefore results natural to generalize this formula by replacing the identity operator $\operatorname{Id}$ with an arbitrary operator between separable Hilbert spaces and $\operatorname{id}$ by a discrete mapping, which is Definition 3.1.
> 13. In a CRV, we will ensure that all figures and diagrams have captions and references within the main text with the necessary explanations.
> 14. The relevant aspect of SNO in terms of the representation equivalence is not so much the specific form of the neural network in Eqn 4.3, but rather the fact that SNO first performs an analysis step by extracting the Fourier coefficients, then the neural network acts only on the coefficients and then, the resulting coefficient vector is synthesized back to a signal that is periodic and bandlimited.
> 15. Yes, Eqn 4.3 accounts for a bias term.
>
> We sincerely hope to have addressed your concerns, particularly about the mathematical details, satisfactorily and would kindly request the reviewer to update their assessment accordingly.

---

> > ### Comment · Reviewer_2w9i · 2023-08-13
> > **Reply**
> >
> > Thank you for the detailed responses to my questions. I have read through all of your answers, and I now have a better understanding of the points that were initially unclear to me.
> >
> > Regarding specific points:
> >
> > * I appreciate your responses regarding the aspects I missed in questions 3, 8, 14, and 15.
> >
> > * It would be beneficial if the sections addressed in questions 1, 2, 7, 9, 10, 11, 12, and 13 could be revised and incorporated into the paper in the future.
> >
> > * I have thoroughly reviewed the additional experiments and explanations for questions 4, 5, and 6. I understand that the ultimate conclusion is that not only SNO but also CNO and PCANet all exhibit an aliasing error of 0, resulting in them being representation equivalent and leading to ReNO. As the primary goal of operator learning models is to excel in operator learning for accurate solution predictions, could you further explain the final implications of these three models being ReNOs? While I comprehend the theoretical analysis and experimental demonstrations, the ultimate question remains: do these findings translate to improved solution predictions in practical applications?
> >
> > * Furthermore, I have examined the CNO paper (https://arxiv.org/pdf/2302.01178.pdf), where the concept of a 'representation equivalent neural operator' is also discussed to showcase CNO's fulfillment of this property. What sets your work apart in this context? I am curious if there have been many prior papers demonstrating and discussing their operator learning models as 'representation equivalent neural operators.'
> >
> > * Lastly, I came across the paper at https://arxiv.org/pdf/2207.10241.pdf, which also appears to focus on fixing the basis while learning coefficients. Could all models that learn only coefficients be considered as ReNOs?
> >
> > I truly appreciate your comprehensive responses, and I apologize for any inconvenience caused by my additional inquiries.

---

> > > ### Author Response · Authors · 2023-08-17
> > > **Reply to the Reviewer**
> > >
> > > At the outset, we thank the reviewer for your prompt feedback on our rebuttal. We take this opportunity to address further questions below.
> > >
> > > 1. We will certainly incorporate the changes that we have outlined in a CRV, if accepted.
> > >
> > > 2. Regarding the reviewer's question about *final implications of these three models being ReNOs* and *do these findings translate to improved solution predictions in practical applications*?, we start by stating that it is not obvious to claim that possessing the ReNO property suffices to ensure improved approximation of operators as aliasing error (which directly corresponds to the ReNO property or lack of it) is just one component of overall error and other components such as training, approximation and generalization errors might also play a role. That being said, we can make an argument for architectures where having the ReNO property does lead to improved performance over not possessing this property. A good example is provided by the CNO model of Raonic et. al. where they present the results of an ablation study in Table 11, Page 56 and compare CNO which is a ReNO with CNO w/o filters in which the filtering operations that make CNO a ReNO are ablated out, resulting in a model that is no longer a ReNO. We see from this table that for every single benchmark, the ReNO version of CNO significantly outperforms its non-ReNO version. Moreover, a *very essential* practical manifestation of the ReNO property is the ability of a ReNO to switch between resolutions without incurring aliasing errors. This feature is of great importance in practical problems where operators need to be evaluated on grids at different resolutions. In this case, just as we show in Figure 4 of our paper and of the uploaded rebuttal pdf, not having the ReNO property can lead to aliasing errors, when the operator is evaluated on different grids. This feature also is observed for realworld data sets in Raonic et. al (Figures 2 and 24), highlighting another implication of the practical importance for possessing the ReNO property.
> > >
> > > 3. Regarding the reviewer's question about *what distinguishes our paper from Raonic et al (CNO paper)*, we would like to point out that our paper proposes the general analytical framework of representation equivalence and derives the concept of aliasing error for operators on which representation equivalence is based. CNO is just one, albeit very powerful, example of ReNO as there are others (SNO, PCAnet, DeepONet etc). The authors of Raonic et. al. do not invent the concept of ReNOs and they themselves make it very clear in their discussion (last paragraph of page 12 of Raonic et al) that they only realize one example of ReNO but do not propose the overall theoretical framework, which is the core contribution or our paper.
> > >
> > > 4. Regarding the reviewer's question about whether *Choi et. al's model is a ReNO*, our preliminary reading of this paper is its core model is a slight perturbation of SNO, but with a Legendre, instead of a Fourier or Chebychev basis and on top of it, the authors train their operator with a PINN-type loss. Given its close connection with SNO, we believe that this model is also a ReNO. In general, if a model only learns coefficients, then it could be a ReNO provided that the right input and output spaces and their corresponding frames are taken into consideration, please see the discussion about DeepONet in Appendix E of our paper.
> > >
> > > We hope that we have addressed the reviewer's questions to your satisfaction and if so, we would request the reviewer to kindly revise your assessment of our paper accordingly.

---

> > > > ### Author Response · Authors · 2023-08-20
> > > > **Requesting the reviewer for feedback**
> > > >
> > > > Due to imminent closure of the discussion period, We kindly request the reviewer to provide us with your valuable feedback on our reply and we are at your disposal to answer any further questions in this regard. We also hope that we have addressed your further questions to your satisfaction and kindly request the reviewer to update their assessment accordingly.

---

> > > > > ### Author Response · Authors · 2023-08-21
> > > > > **Further request to the reviewer**
> > > > >
> > > > > As the discussion period is closing, we sincerely hope that we have addressed the reviewer's follow-up questions to your satisfaction. If so, we kindly request the reviewer to upgrade your assessment of our paper accordingly.

---

> > > > > > ### Comment · Reviewer_2w9i · 2023-08-22
> > > > > >
> > > > > > I appreciate your understanding regarding the delayed response. Your answer has resolved my concerns. I will raise my score. Thank you.

---

> > > > > > > ### Author Response · Authors · 2023-08-22
> > > > > > > **Thanking the reviewer**
> > > > > > >
> > > > > > > We sincerely thank the reviewer for their positive comments and for raising the score of our paper.

---

### Official Review · Reviewer_3pPU · 2023-07-07

**Soundness:** 3 good
**Presentation:** 2 fair
**Contribution:** 3 good
**Rating:** 6
**Confidence:** 2

**Summary:**

In this paper, the authors investigate the concept of neural operators in the context of operator learning architectures. They address the fundamental question of what defines a neural operator and propose the notion of Representation equivalent Neural Operators (ReNOs) that satisfy a systematic consistency between continuous and discrete representations, referred to as continuous-discrete equivalence (CDE).

The authors present a framework for analyzing existing operator learning architectures and determining whether they qualify as ReNOs. They emphasize the importance of enforcing CDE at each layer of the approximating operator to ensure genuine learning of the underlying operator, rather than just a discrete representation. Experimental analysis is conducted to validate the theoretical claims and demonstrate the practical implications of respecting representation equivalence.

**Strengths:**

1. Originality: The paper introduces a novel perspective on neural operators in the context of operator learning architectures by focusing on the concept of continuous-discrete equivalence. The notion of Representation equivalent Neural Operators (ReNOs) provides a fresh and rigorous definition that emphasizes the importance of maintaining consistency between continuous and discrete representations. This original approach extends the understanding of neural operators and adds a new dimension to the analysis of operator learning architectures.

2. Quality and Clarity:  the framework is well-structured, and the definitions and mathematical formulations seem correct and rigorous.

3. Significance: The paper holds significant importance for the theoretical analysis of operator learning architectures.

**Weaknesses:**

1. Experimental Validation: Although the paper includes experimental analysis to support the theoretical claims, there is room for further elaboration and validation, e.g. incorporating more SOTA architectures and more benchmark datasets.
2. Practical Significance: Lack of a principled architecture instance or design. The paper does not propose a new architecture satisfying the ReNO definition, and does not provide a systematic way to design new instances of ReNO.
3. Readability: Lack of intuition and hints between mathematical definition and derivation, especially in section 3.


**Questions:**

1. Is SNO only suitable for periodic signals? If one uses a chebyshev basis instead of fourier basis to implement SNO, then does the analysis in the paper still hold?

**Limitations:**

The authors provide a through and insightful discussion about limitations and extensions in Section 6.

---

> ### Author Rebuttal · Authors · 2023-08-07
>
> We start by thanking the reviewer for their appreciation of the merits of our paper and their welcome suggestions to improve it. We address their detailed concerns below.
>
> 1. Regarding the reviewer's concerns about *experimental limitations, lack of SOTA architectures, benchmarks*, we start by saying that the main point of our paper was to provide a novel theoretical framework of representation equivalence, based on a non-trivial and novel definition of aliasing errors for general operators. In our view, this is the crucial contribution of this paper and we apply it to analyze whether and when existing neural operator architectures are ReNOs. In the paper, we have already analyzed CNN, FNO (not ReNOs in general) and SNO, DeepONet (in SM Sec. E) which are ReNOs. Following your and other reviewer's suggestions, we have also analyzed the PCA-net architecture of Bhattacharya et al as a ReNO (see general response to all reviewers as well as the attached 1-page pdf). We would also like to mention that recently, CNO (Arxiv:2302.01178) was also shown to be a ReNO. The authors of CNO show that this ReNO is state-of-the-art on learning operators corresponding to a variety of PDEs, already demonstrating the practical utility of the concept of ReNO. We will add further discussion about these architectures in a CRV, if accepted. Moreover, we have reported additional experiments in the 1-page pdf and described them in the general response to all reviewers. In particular, we show how FNO can be made into a ReNO (approximately) by minimizing the differentiable aliasing error (Eqn 5.1 of the main paper).
>
> 2. Regarding the reviewer's comment about a *systematic way to design ReNOs*, in addition to the reply in pt 1, we would like to emphasize that one of our key contributions is Eqn (3.6), which specifies how ReNOs act under a change of frame sequences. This is novel and allows for a principled way in which resolution can be varied for a ReNO and it has been utilized in the design/evaluation of CNO for instance. Moreover, we have added a possible algorithm on how non-ReNO can be made into a ReNO by minimizing aliasing error, without any significant loss of expressive power -- please check the general response to all reviewers as well Figure 2 of the attached 1-page pdf.
>
> 3. Following the reviewer's excellent suggestion,  we will include more intuition and hints regarding the mathematical aspects of Sections 2 and 3 in the CRV, if accepted. As an example, we plan to add a cartoon depiction of the ReNO framework  that we have created in Figure 1 in the uploaded pdf for this rebuttal. Hopefully, this will make our underlying concepts clearer to a reader.
>
> 4. The work on SNO (Fanaskov et al) indeed already considers both choices of basis functions: Chebyshev and trigonometric polynomials. Thus, SNO is a ReNO with respect to a Chebyshev basis too.
>
> We sincerely hope to have addressed your concerns, particularly about the practical utility and realization of ReNOs, satisfactorily and would kindly request the reviewer to update their assessment accordingly.

---

> > ### Comment · Reviewer_3pPU · 2023-08-16
> >
> > Thank you for the response, my questions have been well addressed.

---

> > > ### Author Response · Authors · 2023-08-17
> > > **Thanking the Reviewer**
> > >
> > > We thank the reviewer for your reply and are at your disposal if you have any further questions during the discussion period.

---

### Official Review · Reviewer_9wa4 · 2023-07-11

**Soundness:** 3 good
**Presentation:** 3 good
**Contribution:** 3 good
**Rating:** 6
**Confidence:** 4

**Summary:**

This paper investigate the aliasing error of operator learning. The work points out that many existing neural operators have aliasing error, so the error is higher on super-resolution problem. To deal with the aliasing error, the paper proposed Representation equivalent Neural Operators (ReNO), which have no aliasing error. Numerical experiments show ReNO model perform better with super-resolution task.

**Strengths:**

The paper investigates a very interesting problem -- the concept of resolution invariance in operator learning. It introduces frame theory to study the aliasing error in operator learning and it proposes a neural class of model called Representation equivalent Neural Operators (ReNO). The paper helps the community to better understand the key concept of neural operator.

**Weaknesses:**

I think it's very important to distinguish the two different concepts:

### Discretization invariance vs representation equivalence
- Previous work define neural operator as asymptotic discretization invariant model that converge as the grid refines [1]. In a sense that the neural operator model will converge to the underlying solution operator in continuum. In the case of Fourier representation, discretization invariant FNO model has the capacity to learning infinite number of Fourier basis as created by the non-linear activation layer, which in the end, will cause the aliasing error.

- In this work, the authors the representation equivalent model that has no aliasing error. It is equivalent to say these models are prescribed in a fixed linear subspace of the underlying infinite-dimensinal function space. In the case of Fourier representation, the representation equivalent SNO model has a fixed number of Fourier basis, and a consequential irreducible approximation error.

While the newly proposed representation equivalent models have no aliasing error, they have a limited expressive power, which prevents the model extrapolates to unseen models and higher frequencies. Since the ReNO models cannot use non-linear activation layers after the inverse spectral transform, the output space is restricted to the spectral space, and the model is of the class of linear reconstruction operator [2], which may require exponentially more number of parameter compared to non-linear reconstruction operator like FNO.

In practice, the ReNO models probably perform better at super-resolution, while previous asymptotic discretization invariant models perform better at fixed resolution or mixed resolution training.

### Writing
Besides, the overall writing is very clear. The reviewer kindly suggests to use a more informative title.


[1] Kovachki, Nikola, et al. "Neural operator: Learning maps between function spaces with applications to PDEs." Journal of Machine Learning Research 24.89 (2023): 1-97.
[2] Lanthaler, Samuel, et al. "Nonlinear reconstruction for operator learning of pdes with discontinuities." arXiv preprint arXiv:2210.01074 (2022).



**Questions:**

The CNO work [1] describes a method that makes neural operator model using activation layers also aliasing-free. Specifically it apply upsampling and downsampling before and after band-limited activation layers. As claimed in Proposition 2.1, CNO are representation equivalent.

I wonder if other works such as FNO enjoy a similar properties. Since the IFFT is equivalent to upsampling and FFT is equivalent to downsampling. FNO can restrict the number of Fourier modes of its representation space too if equipped with a band-limited activation function. On the Fourier space FNO truncates the latent function to N Fourier modes, and the band-limited activation function introduce at most M new Fourier modes, so the highest number of Fourier modes is N*M. If N*M < the resolution, there will be no aliasing error.

If it is the case, does FNO also belong to ReNO?


[1] Raonić, Bogdan, et al. "Convolutional Neural Operators." arXiv preprint arXiv:2302.01178 (2023).

**Limitations:**

If will be better to discuss the relationship between the aliasing error and approximation power.

---

> ### Author Rebuttal · Authors · 2023-08-07
>
> We start by thanking the reviewer for their appreciation of the merits of our paper and their welcome suggestions to improve it. We address their detailed concerns below.
>
> 1. Following the reviewer's welcome suggestion, we propose to change the title to the hopefully more informative *Representation equivalent Neural Operators: Frame Theory Meets Operator Learning*.
>
> 2. On the reviewer's question about *link between discretization invariance (DI) and representation equivalence (RE)*, we start by pointing out that these concepts are indeed different and to some extent, complementary. This difference is precisely what motivated the current paper. We believe that, in practice, the main utility of the  notion of representation equivalence is that it precisely describes what is happening when changing resolutions, when compared to discretization invariance which corresponds to a notion of asymptotic consistency in the infinite resolution limit. Using the notion of RE, we can interpret varying resolution plots, such as those in Kovachki et al or even in Fig. 4 of our paper, while the asymptotic consistency requirements of DI do not explain why the observations of these figures hold true as one cannot rule out large errors between resolutions when consistency is only required in the infinite resolution limit. Given this context, we would provide a detailed discussion of DI vs. RE, emphasizing their complementary roles in a CRV, if accepted.
>
> 3. Regarding the reviewer's questions about *linear vs non-linear methods in the context of operator learning*, we would like to start by clarifying that the results of Lanthaler et. al. were specific to the case of scalar 1-D transport of a single discontinuity and a related 1-D Burgers example where they claimed that FNO is better than DeepONet as the approximation error of a linear reconstruction operator can be bounded below by the rate of decay of eigenvalues that may converge slowly for some transport equations. This does not imply that methods with a linear reconstruction cannot be efficient or have limited expressive power for a very large class of PDEs, say elliptic and parabolic PDEs, which have a regularizing effect resulting in fast spectral decay. Moreover, the result of  Lanthaler et. al. only pertains to size efficiency vis a vis approximation error. In the case where model size $\ll$ resolution of the data, the computational bottleneck is data resolution, which we have no control over, as it is given to us a-priori. In this case, using a linear method does not necessarily increase complexity as the data resolution dominates. Moreover, other sources of error, such as aliasing error, discussed extensively in our paper can also affect the performance. Finally, as the reviewer has pointed out the work of Raonic et. al. on CNO -- Table 1 in Raonic et. al. shows that a ReNO like CNO can outperform FNO even on benchmarks with discontinuities (Transport Eqn.) and Shocks (Compressible Euler Equations), attesting to the abilities of methods based on linear reconstruction for approximating operators with slow spectral decay.
>
> 4. In reply to the reviewer's excellent question, it is indeed possible to modify the FNO architecture in a similar spirit as it is done in CNO of Raonic et al. , to respect the continuous-discrete equivalence. We will mention this fact in the CRV, if accepted. Here, we have explored another option for making FNO into a ReNO, which we have described in some detail in our general response to all reviewers above. Our approach consists of training FNO to simultaneously minimize the regression error as well as the discrete aliasing error (Eqn. 5.1 in our paper). The results in Figure 2 of the attached pdf show that the aliasing error is indeed minimized by this approach while the training error remains as small. This experiment shows that working directly to minimize the aliasing error is also a viable approach for making a neural operator into a ReNO.
>
> 5. Regarding your comment about *link between approximation and aliasing errors*, we would like to point out that there is no intuition to believe that there is a possible link (or even trade-off) between aliasing error and approximation power. Moreover, the experiment on FNO (see pt. 4 above) reported in Figure 2 in the uploaded pdf for this rebuttal further strengthens the argument that such a link does not exist. It is rather the case that aliasing error is an artifact that can be eliminated without necessarily losing approximation power. Following your suggestion, we will include a detailed discussion of this issue in the CRV, if accepted.
>
> We sincerely hope to have addressed your concerns, particularly about the differences between discretization invariance and representation equivalence, satisfactorily and would kindly request the reviewer to update their assessment accordingly.

---

> > ### Author Response · Authors · 2023-08-19
> > **Requesting the reviewer for feedback**
> >
> > Due to imminent closure of the discussion period, We kindly request the reviewer to provide us with your valuable feedback on our rebuttal and we are at your disposal to answer any further questions in this regard.

---

### Official Review · Reviewer_nALh · 2023-07-26

**Soundness:** 3 good
**Presentation:** 3 good
**Contribution:** 3 good
**Rating:** 6
**Confidence:** 5

**Summary:**

The concept of “representation equivalence” is introduced in the context of operator learning. The definition amounts to requiring zero aliasing error from the mode. It is shown whether several popular architectures satisfy the new definition.

**Strengths:**

The paper is well written and easy to follow. The math is made accessible and good intuition is given. Claims are proven rigorously with detailed proofs.

**Weaknesses:**

While I like the general direction of this paper, I do not appreciate catchy titles that obfuscate mathematical ideas for the purposes of a “wow” factor. Are Neural Operators Really Neural Operators? Yes they are because that’s how they were defined in [14]. Putting titles aside, I think that two ideas about what should constitute “operator learning” are being mixed. The point of [14] is to define families of architectures whose parameters are not tied to the discretization of the input or output functions. This makes the cost of a model (with cost defined as the size of the model) independent of discretization. The analogy is to explicit vs. implicit methods for solving time-dependent PDEs. For an explicit method the cost (defined here as the inverse size of the time step) increases as the spatial discretization increases due to CFL. On the other hand, for implicit methods, the time step and the spatial discretization are independent. From this point of view, neural operators are analogous to implicit methods while, say CNNs, are analogous to explicit methods. This makes CNNs a perfectly reasonable method for operator learning, however, the challenge which remains is how does one increase the size of the CNN with the resolution in order to guarantee consistent results. This can then map into a proper cost-accuracy trade-off analysis between methods just as one can do for explicit vs implicit methods. Note that for this to be done, architectures need to be re-trained at each testing resolution. Indeed what is shown in [14] is if one re-trains the same model with higher and higher resolutions, the answer remains consistent. A consequence of the neural operator definition (parameters not tied to grid points) is that a trained architecture can be used to predict at multiple resolutions, however, one cannot, in general, have guarantees that the answer will stay consistent. Indeed if the true operator transforms all modes of an input function in some way then how can one guarantee that a surrogate trained only on some of the modes will generalize to unseen modes? One can hope that the surrogate extrapolates correctly and that has empirically been shown on some problems (some in [14] but also many others), but it cannot be guaranteed any more than generalization abilities of deep neural networks can be guaranteed. From this point of view, neural operators do precisely what they are meant to do,
and I strongly think that the authors should be much more clear and explicit about the problem that their work tries to address because it is ultimately a different one.

I like the idea of quantifying and mitigating the aliasing error. It is a very important direction that has been almost unexplored in deep learning with many practical consequences. However, I do think that Definition 3.4 is much too strong to be useful. Indeed, in Remark 3.5, the authors show that, as a consequence of this definition, there is only one recipe for constructing ReNO architectures, mainly eq. (3.6). Note the similarity of this equation to eq. (2.3), upon combining with eq. (2.2), in https://arxiv.org/pdf/2005.03180.pdf. I point this out because constructions of this kind yield only linear methods of approximation. It is well known in function approximation that linear methods are suboptimal (for example, https://people.math.sc.edu/devore/publications/NLACTA.pdf). While this has certainly not been studied nearly as deeply in the operator setting, similar empirical and theoretical results are beginning to emerge (for example, https://arxiv.org/pdf/2210.01074.pdf). I therefore struggle to see the practicality of such a strong definition. For what kinds of problems would this be useful?

Lastly, I think the numerical experiments are quite insufficient in demonstrating the practical usability of ReNO. Is aliasing error actually prevalent or is the main source of error, the approximation error? If the latter, should architecture design not be focused on reducing approximation as opposed to aliasing error? What is the trade-off between approximation and aliasing, and can an architecture which is not ReNO, still be modified to reduce aliasing error while retaining the same approximation power?

**Questions:**

I do not understand the details of the numerical example. After having read the paper and the Appendix, I feel that I am missing something. I will describe the set-up to the best of my knowledge. I will denote by f the input function and by u the output i.e. Q(f)=u. For simplicity, consider only two resolutions and denote by x_1,...,x_m the first discretization and by y_1,...,y_n the second with the assumption that they are nested so that {x_j} is a proper subset of {y_j}. Data is generated by randomly (from a Gaussian) sampling the point values f(y_j) and u(y_j). Three neural network based methods are then trained to regress on samples f(x_j) and u(x_j). They are all then tested on fitting u(y_j) from f(y_j) (I assume with samples from a test set). It is shown that the SNO achieves the same error on regressing u(x_j) from f(x_j) as it does on regressing u(y_j) from f(y_j) while the other two methods do not. I do not understand how this is possible. The SNO approximates Q in the form Q(f)(x) = sum_{k=-d/2}^{d/2-1} g_k ({<f, e^{i .}>, …, <f, e^{id .}>}) e^{ikx} where g is a neural network mapping C^d to C^d with d some fixed positive integer. By passing from f(x_j) to f(y_j), the input to g changes since the inner products now see f(y_j),  but the periodic bases stay the same. The values of u(y_j) were picked at random and independently from those of f(y_j) so how can this new information about f possibly have an effect on the prediction of u? In general, this does not seem possible since whenever Q transforms all modes of f whatever method that regresses Q must extrapolate this transformation to new modes. While this may happen sometimes, it cannot be guaranteed. But this looks completely impossible in the current experiment since the SNO must extrapolate randomness. There must be something in this experiment that I am missing; it would be helpful for the authors to clarify.

While the authors do not propose the SNO, it is the only method which they show satisfies ReNO, so I do wonder about it. It is well known that, for approximation in L^2, an optimal linear subspace is given by PCA. This is the motivation behind PCA-Net [3] which is precisely like the SNO but, instead of picking Fourier bases, computes bases for the inputs and outputs by doing PCA on the data. What is the motivation behind sticking with Fourier bases? In fact, since any method which is ReNO is a linear method of approximation, is PCA-Net not the optimal ReNO method (barring optimality of the finite-dimensional neural network)? Generally, if the ReNO definition is to be adopted as a standard way of doing operator learning, how do the authors envision new architectural designs? Remark 3.5 fully characterizes ReNO architectures, so it is hard for me to see a way forward.

The first example of Section 4 shows that discrete convolution layers are not ReNO, yet the work https://arxiv.org/pdf/2302.01178.pdf (CNO, Proposition 2.1) claims that they are. As far as I can tell, the way that ReNO is proved for CNO is by showing that discrete convolution maps w-bandlimited functions to w-bandlimited functions and then constructing a pointwise nonlinearity that does the same. I am not convinced by the CNO proof since nonlinearities used in practice (like ReLU or, more generally, learned MLPs) will not map bandlimited functions to bandlimited functions of any order and, so, even going through the upsampling-nonlinearity-downsampling procedure, will not give an aliasing error that is exactly zero (but it will mitigate the aliasing error as was first proposed in https://arxiv.org/pdf/2106.12423.pdf). Putting nonlinearities aside, however, one still needs that the discrete convolution is ReNO, so where do the differences lie?

How do the authors envision a similar theory for Banach spaces when the $\ell_2$ isomorphism breaks down?

**Limitations:**

The authors adequately address potential negative societal impact.

---

> ### Author Rebuttal · Authors · 2023-08-08
>
> We thank the reviewer for their constructive comments and remarks, which we address below:
>
> 1. We strongly acknowledge the significance of neural operators, as defined in [14] as a concept, and this is exactly why we intend to draw inspiration from it and further develop upon it. Although the reviewer may have interpreted our approach as a challenge to the Neural Operator (NO) concept, this was never our intention. To the contrary, our research aims to advance better understanding of various aspects of NOs. For this reason and to avoid any possible misunderstanding by intended readers, we propose to change the title to a more informative "Representation equivalent Neural Operators: Frame Theory Meets Operator Learning".
> 2. With this context, one of our goals was to analyze under what conditions do NOs actually correspond to the mathematical notion of operators. Indeed, NOs satisfying just the discretization invariance property may not necessarily correspond to operators, leading to possible negative implications such as loss of *structures* like symmetries and locality, only defined at the continuous level, leading to possibly poor generalization as well as inconsistencies across resolutions. Our aim here was to develop a theoretical framework that effectively addresses these aspects. We demonstrate that when seeking to establish a relationship between continuous and discrete or between different resolutions, the concept of aliasing is the crux and as acknowledged by the reviewer, one key and highly non-trivial contribution of our paper was a rigorous definition of aliasing errors for operators (Eqn (3.1)) which is the first instance of this formula in the literature. Moreover, we believe that aliasing complements the consistency errors for NOs in [14] very well. We will highlight this contribution more prominently in the CRV, if accepted.
> 3. Regarding the *rigidity and utility of Def 3.4*, as the reviewer suggests, it automatically leads to the possibility of not necessarily setting the aliasing error to zero, but to control it and make it as small as desired. This is a natural corollary of def 3.4 and we will discuss it at greater length in the CRV (see also pt below). At the same time, we politely disagree with the reviewer's contention that Def3.4 (Rem 3.5) is too restrictive. It leaves the room to choose frames at each layer, as well as the neural net model itself, i.e., nonlinear layer $G_\ell$. Choosing the frames $\Psi_\ell$ and $\Psi\_{\ell+1}$ for the input and output spaces, only specifies how the corresponding discrete operator $g_\ell(\Psi_\ell,\Psi\_{\ell+1})$ has to be chosen. This just states that moving from continuous to discrete representations has to happen in a precise and controlled manner, but does not specify the architecture. The neural operators $G_\ell$ are to be learned and are not fixed by the ReNO framework and there is considerable room for innovative architectural designs here.
> 4. Regarding *linear vs. nonlinear models in operator learning*, we start with the contention that frames based on multiresolution, such as wavelets, allow in fact to go beyond linear approximation of signals to *nonlinear approximation* (in the sense of DeVore, Acta Num 1998). With such frames, one can extract signal-adapted approximations that can zoom in where necessary, instead of using a uniform discretization. Explicitly implementing such constructions are topics for future research, but our framework of considering general frames is, in our opinion, the right setup to look for such architectures. Moreover in this context, the reviewer defines the cost as model size. We believe that the overall computational complexity is more pertinent. This is a subtle but important point. In the case where model size $\ll$ resolution of the data, the computational bottleneck is data resolution, which we have no control over, as it is given to us a-priori. In this case, using a linear method does not necessarily increase complexity, putting into question the utility of so-called *non-linear reconstruction* operators of Lanthaler et al. Clearly, more discussion on this point is necessary and will be included in a CRV, if accepted.
> 5. We apologize for the possible lack of clarity in our description of the numerical experiment in Sec. 5. We have added a detailed description in the general response to all reviewers above and request the reviewer to examine it and ask us further questions, if needed.
> 6. The reviewer's excellent questions about *trade-off between approximation and aliasing errors* and *can a non-ReNO NO be made to reduce aliasing* inspired us to perform the follow-up experiment described in the general response. As shown in figure 2 of the attached pdf, we were able to reduce the aliasing error of FNO significantly by minimizing the discrete aliasing error (Eqn 5.1) (Fig.2 Left) while keeping training errors small (Fig. 2 Right). In this example, there appears to be no trade-off between aliasing and approximation errors!
> 7. SNO is simply one example of a ReNO and we have analyzed when DeepONet is a ReNO (SM E, lines 480-489). Following the reviewer's suggestion, we can show that PCANet is also a ReNO, see Fig 3 (attached pdf) for the commutative diagram and Fig. 4 for experimental verification. However, we also observe that the resolution invariance of PCAnet is very sensitive to slight perturbations like randomly removing even one data sample while evaluating on a different resolution (See plot for PCAnetJitter in Fig. 4). In the CRV, we will investigate this issue carefully, particularly in the context of the question about *optimality* of PCAnet.
> 8. It is indeed possible, although technical, to derive a ReNO framework in Banach spaces using Banach frames (Casazza et. al, 1999). We will add a discussion on this topic in the CRV.
>
> We sincerely hope to have addressed your concerns satisfactorily and would kindly request the reviewer to update their assessment accordingly.

---

> > ### Comment · Reviewer_nALh · 2023-08-11
> > **Points 1-4**
> >
> > Thank you to authors for the detailed response and added clarifications. I'll respond to each point separately:
> >
> > 1. Thank you. I like the new title.
> >
> > 2. I agree that there can be a loss of structures when moving from the continuous the discrete. This can, and does, happen in numerical methods as well; for example, numerical dissipation can occur for operators that are known to be non-dissipative in the continuum. I am far from convinced that aliasing error is at "crux" of this issue however. I think this should be addressed in a problem dependent way based on what is known about the operator of interest; there is no "one fits all" solution. In fact, the solution presented here preserves no structure of the operator to be learned but rather proposes consistency with respect to a predefined frame that is independent of the solution operator. Consider an example. Since basically everything done in practice deals with pointwise evaluations, we can take as the "right" frame, the one defined by the sinc functions. So any ReNO architecture should essentially upsample by a sinc filter. Take Navier-Stokes (NS) as the operator of interest. If I introduce more fine scales in the initial condition (even just by sinc interpolation), I will end up with a solution which has a finer scale turbulent structure. Surely if I take a low resolution NS simulation and upsample it by a sinc filter, I will not end up with the high resolution simulation solution. Yet a ReNO is constrained to do so. In fact, this remains true of the input data as well. Take a microstructure problem where the material interface is discontinuous. A sinc interpolation will not yield the right boundary structure. Even if I take as input the samples from a Gaussian field then higher resolution samples correspond to a particular scaling of the high frequency modes based on the eigenvalues of the covariance. These have nothing to do with the sinc interpolation consistency being imposed here. In practice, why should I try to do "super-resolution" with a ReNO operator when I can just as well take my low resolution solutions and upsample with a sinc filter?
> >
> > 3. Right but this not currently in the paper. As written, a model is ReNO if and only if the aliasing error is zero and what is shown is that some models satisfy this while others do not. This can be quite misleading for readers as it seems to make the implication that some models do "true" operator learning while others do not. Furthermore, I am generally confused why it is imposed that every layer in an architecture needs to have zero aliasing error. At the end of the day, if you want zero aliasing error then it enough to simply make the last layer have the form (3.6). Am I missing something?
> >
> > 4. Linear vs. non-linear has nothing to do with wavelets or any particular choice of frame/dictionary. The question is what is the structure of the range space of the learned operator. If the range space is a linear subspace of the Hilbert space then it is a linear method and results concerning the Kolmogorov n-width apply (as pointed out in Lanthaler et al.). Based on definition 3.4, the range space of any ReNO operator is the span of a fixed, finite number of functions belonging to the frame \Psi_L which is therefore a linear subspace. A possible way of making a non-linear architecture is to adaptively pick, based on the input function to the operator, which functions from \Psi_L are used in the final projection. However, as I understand it, definition 3.4 does not allow for this as the synthesis operator is fixed. Furthermore, I am not sure how this will interact with definition 3.1 which defines aliasing relative to an operator rather than a function. There are other ways of making non-linear architectures, for example, applying a learned (or fixed) Nemitskii operator to the output of the last layer. This is how the FNO is made non-linear. But, again definition 3.4 does not allow for this. Another possibility to is make the last layer a non-linear integral kernel transform, for example the attention layer in transformers, but again, the definition rules this possibility out. I am not sure I understand the point about model size. While I agree that computational complexity is more important than number of parameters, the two are corrolated and one can derive exact formulas as done in https://arxiv.org/abs/2203.13181. The number of parameters in most models used in practice is very large. Even the smallest models usually have at least a million parameters; this correspond to data, in 2d, given at a 1000x1000 resolution which seems quite high. I am not sure what is meant by "putting in question the utility" of the methods from Lanthaler et al. Is the implication that if data in the experiments performed there was given at a much much higher resolution, the linear methods would be competitive with the non-linear ones? I have a hard time believing this.

---

> > > ### Comment · Reviewer_nALh · 2023-08-11
> > > **Points 5-9**
> > >
> > > 5. Thank you. I understand the experiment now. I think it would very much improve readability if this is put into the main part of the paper.
> > >
> > > 6. This is a great experiment and I am grateful that the authors are thinking of possible ways of reducing the aliasing error that are applicable to a broad class of operator learning architectures. My concern here, however, remains the same as what I pointed out in 2. The experiment only considers training error and not generalization error. What would be the impact of this new loss function on real data where the true operator of interest is not consistent with sinc interpolation? My intuition says that it would hurt the overall performance. Generally, I think it is really important to demonstrate the benefits of the proposed framework on a real dataset and not just a synthetic example.
> > >
> > > 7. Thank you for this. I suspect that the "jittering" has to do with numerical instability of the algorithm used to compute the PCA modes since the data is random. If smoother data is used, I expect this to disappear. For optimality, I suggest https://arxiv.org/abs/2303.16317.
> > >
> > >  8. Thank you for the nice reference.
> > >
> > >  9. I remain curious about the discrepancy in proofs concerning the ReNO property of the discrete convolution layer given in the present paper and in https://arxiv.org/abs/2302.01178. Would be great if the authors can clarify.

---

> > > > ### Author Response · Authors · 2023-08-18
> > > > **Reply to the Reviewer's comments: Part I**
> > > >
> > > > At the outset we would like to thank the reviewer for the detailed and prompt feedback as well as follow-up questions. Before answering them, we wish to reiterate that our article is aimed at complimenting existing notions of neural operators by highlighting what we believe is a very important issue i.e., continuous-discrete equivalence and aliasing errors which accrue when it is violated. We genuinely believe that our insights and findings have the potential to contribute positively to the community working on operator learning. In response to your very detailed and valuable feedback, we will refine our presentation of the aliasing error,  our exposition of the main ideas, as well as modify the title. Given this is the first work on the topic, it can be somewhat challenging to communicate the main ideas within the strict page-limit and we will try our best to do so in a CRV, if accepted. With these remarks, we answer your detailed questions below and thank you in advance for your patience in reading our reply:
> > > >
> > > > **Reply to Qn (2)** *Discretizations of ReNO*: Traditional numerical methods directly approximate the solution operator as they have access to the underlying differential equations. In contrast, our focus with ReNO is not on discretizing the solution operator directly, but rather at correctly discretizing the Neural Operator (NO), defined at the continuous level, that seeks to approximate the underlying solution operator. While this might appear as a nuanced distinction, we deem it crucial. Indeed, the Neural Operator does not have access to the solution operator nor to the underlying equation, only to data, mostly derived from numerical solutions. Thus, for the example cited by the reviewer, we find it hard to believe that any NO can discern high-frequency content given the numerical solution, on which it is trained, is itself dissipating, and that there is a limit to the frequencies the data can exhibit. Given this rationale, we propose a pragmatic approach, i.e., first consider a (Representation-Equivalent) Neural Operator between function spaces that are in correspondence with the finite resolution data (zero, or small aliasing), for instance, generated by the continuous counterpart of the dissipating numerical system. The second stage--and this is where the solution operator comes in-- is the analysis of whether the numerical solution approximates the true solution operator.  Of course, we expect to suffer from discretizations errors, but we also claim that these errors can be out of reach, assuming the data that is given to us. Thus, our approach is different from the classical Neural Operator approach, and this will be highlighted in the main paper, based on the very good points raised by the reviewer. Moreover, we believe that it is the design of the ReNO (input and range functions) that should reflect properties associated to the numerical solution, and that its discretization (choice of frames) should be chosen in such a way that reflects the continuous ReNO operator (zero, or low and controlled aliasing error).
> > > >
> > > > *Loss of continuous structures*: Our methodology aims to maintain the structure of the Neural Operator (defined at the continuous level) and not necessarily that of the solution operator. Instead of striving for consistency based on a singular predefined basis or frame, our approach embraces multiple bases. As evidence, we revisited the FNO experiment presented in detail in the rebuttal. By retraining the FNO without inputting the grid, the primary structure we aim to uphold is translation equivariance, defined at the operator level (given that translation is continuous). As we adjust the parameter $\lambda$ weighing the aliasing error, there is a notable correspondence between the translation equivariance error and the aliasing error.
> > > >
> > > >
> > > > |  | $\lambda$=1.0 | $\lambda$=0.2 | $\lambda$=0.1 | $\lambda$=0.0 |
> > > > |---------------------------------------|-------|-------|-------|-------|
> > > > | Discrete Aliasing Error (DAE) (%)  | 10.38 | 20.24 | 42.97 | 98.55 |
> > > > | Translation Equivariance Error (%) | 10.54 | 20.54 | 45.79 | 104.44|
> > > >
> > > >
> > > > This clearly illustrates that (1) aliasing can cause loss of continuous structures (2) these can be recovered by our framework. We are happy to include a further discussion on this issue in the SM of CRV, if accepted.
> > > >
> > > > (Contd.......................................................................................................)

---

> > > > > ### Author Response · Authors · 2023-08-18
> > > > > **Reply to Reviewer's comments: Part II**
> > > > >
> > > > > **Reply to Qn (2) [Contd..]** *Learning Navier-Stokes at different resolutions*: With regards to your example, we would like to point out that the ReNO framework does not only amount to upsampling to obtain the operator related to the Navier-Stokes at fine scales. What it does tell us is that as the numerical solutions at different scales can be inherently different at the continuous level (due to numerical diffusion which is resolution dependant), learning a unique Neural Operator for all scales could be impossible without aliasing. What we would do in this context would be to instead consider a ReNO for each resolution, and sharing parameters across resolutions. This can be achieved with operators like FNO or CNO by taking resolution as input, which is an indeed a very interesting avenue of research. This way continuous operators will preserve structures at each resolution, and we will be able to learn useful representations across resolutions. This is indeed an important problem which we wish eventually to tackle in a follow up work.
> > > > >
> > > > > However, we disagree that everything in practice deals with point-wise evaluations: there are numerous useful applications that require going beyond point-wise evaluations, such as  magnetic resonance imaging, coherent diffraction imaging, or noise reduction in signal transmission, just to name a few. Another very interesting avenue is to go beyond representations by point samples and the space of bandlimited functions (at the price of errors which can be controlled and reduced) to representations in other shift-invariant spaces such as the space of cubic splines. This is achieved by considering pre- and post-filtering, as explained in [Unser Sampling-50 years after Shannon]
> > > > >
> > > > > **Reply to Qn (3)**: We agree with the reviewer that the current version of the paper mostly focuses on the case of zero aliasing error. We had already conceded in the rebuttal that, following the reviewer's feedback, we will also highlight more explicitly the role of the aliasing error itself in a CRV, if accepted. Regarding your question about why the condition of zero-aliasing error is not imposed on the last layer but on every layer, it is essential to emphasize that only imposing the condition at the last layer may not preserve representation equivalence as aliasing errors, with respect to the underlying continuous operator, could have accumulated in intermediate layers and all that imposing this condition on the last layer will ensure is that there is no aliasing vis-à-vis the output of the *last but one* layer, but not for the entire operator (defined as a composition of all layers). For instance, in the case of the FNO (assuming that the activation function is a map between bandlimited functions), the discrete computations may no longer correspond to the continuous form of FNO. Moreover, imposing the ReNO condition layerwise is sufficient to ensure that the composition is alias-free. However, it is not a necessary condition and a multi-layer neural operator could be alias-free, even if this condition is not imposed layerwise.
> > > > >
> > > > > **Reply to Qn. (4)**: *Linear vs. Nonlinear*: We start this reply by clarify that we meant in our rebuttal regarding linear vs. nonlinear approximation. In our understanding of the approximation theory literature (for instance summarized in [DeVore  Nonlinear approximation]) this difference can be described as follows: for a given frame, a linear $n$-term approximation is one that is based on a fixed subset of $n$ indices. A nonlinear approximation is based on choosing the $n$ coefficients of largest magnitude. This set will be different for each function and thus, nonlinear approximations are adaptive. The approximation rates one can obtain depend on the signal class considered as well as on the chosen frame. In this context, we refer to [Vetterli Wavelets, Approximation and Compression] for a nice explanation of the shortcoming of Fourier bases vs wavelet bases for signals with discontinuities. The essential point is that for a Fourier basis, there is no improvement in the approximation rate when transitioning from linear to nonlinear approximation. In two dimensions there are choices that lead to better (nonlinear) approximation rates, see for example [Grohs et. al  Parabolic Molecules: Curvelets, Shearlets, and
> > > > > Beyond]. As mentioned in our previous response, item 4, it is an interesting and natural direction for future work to consider such adaptive and learnable choices of frame coefficients in each layer. This is really just a natural extension of the current ReNO framework. In such a setup, the obtained operator would also be a *nonlinear method* in the sense of Lanthaler et. al, while adhering to the ReNO property. We will further elaborate on this aspect and outline this extension in the discussion section of the CRV, if accepted.
> > > > >
> > > > > (Contd.............................................................................................................)

---

> > > > > > ### Author Response · Authors · 2023-08-18
> > > > > > **Reply to Reviewer's comments: Part III**
> > > > > >
> > > > > > **Reply to Qn (4) [Contd...]** *Our stand on Lanthaler et. al* : As the reviewer frequently refers to Lanthaler et. al, we would like to elaborate how we relate our results to this paper. In our reading, Lanthaler et al compare DeepONet and FNO, with DeepONet as an example of an operator learning method with *linear reconstruction* and FNO an example with *non-linear reconstruction*. With this setup, Lanthaler et. al, proves a lower-bound on the linear reconstruction error and hence the approximation error in-terms of the decay of eigenvalues of the underlying covariance operator. Then for problems such as transport equation where this decay can slow, this implies certain bounds on the model size for DeepONets. Although we did not find any mention of Kolmogorov N-widths in Lanthaler et al, we presume that this relation can be made using the obtained lower bounds. On the other hand, for very simple examples of **scalar 1-D transport of a single discontinuity** and **1-D Burgers' with a sine wave initial data**, the authors were able to show that FNO has better complexity vis a vis DeepONet. Several remarks come to mind immediately 1. There are plenty of problems (corresponding to elliptic and parabolic PDEs) where the eigenvalues decay very fast and this lower bound is moot, 2. The complexity analysis for FNO only holds for very simple situations as described above and it is not at all clear, if this carries over to more general transport-dominated problems 3. CNN is also a non-linear reconstruction method in the sense of Lanthaler et al as is just a pointwise MLP (FNO with Fourier layers switched off) -- just having a nonlinear reconstruction clearly does not guarantee that the method will be superior to one with a linear reconstruction 4. The numerical experiments of CNO (Raonic et. al.) show that a method with a sinc basis (CNO) is quite competitive on discontinuous transport problems (Table 1) vis a vis FNO highlighting that other sources of error might also be relevant and not just the approximation error 5. All traditional numerical methods (Finite element, Finite Volume, Spectral) are based on a fixed basis and the nonlinearity comes in while considering how the coefficients are obtained. Given the above considerations, we agree that a clearer statement needs to be made in the CRV, if accepted, about how our framework relates to the results of Lanthaler et. al, with a careful discussion of the issues that naturally arise in this context.
> > > > > >
> > > > > > Regarding our point about **model size and data resolution**, we would like to clarify and reemphasize our contention in the rebuttal that the overall error will be dominated by the errors that are present in the data. Let us elaborate this point: the results of Lanthaler et. al require access to the exact solution operator. However, in practice, one only has access to numerical approximations of the true solution. These numerical approximations are performed with traditional numerical methods, which have limited accuracy when resolving discontinuities (at most first-order accurate in right norms) -- so in practice, the error of the data resolution will dominate the overall error and possibly render the results of optimal model size inoperative. As an example, consider the linear transport of the single moving discontinuity. Instead of having access to the exact solution, we only have access to data (on a fixed resolution of $N$ points) generated by say a finite difference method. Then by the classical results of Engquist and Sjogren, the discretization error is $\mathcal{O}(1/N)$. Even if FNO learns this data with $\log$-complexity in model-size $M \sim log(N)$ (as shown for the exact solution operator in Lanthaler et. al), the overall error will still be  $\mathcal{O}(1/N)$ , dominated by the discretization error. On the other hand, for DeepONet, even if model size $M \sim N$, the overall error will be $\mathcal{O}(1/N)$  (possibly with a larger constant than the FNO). Finally, the numerical examples of Lanthaler et al rarely consider model size of $M \sim log(N)$ as this would be $\mathcal{O}(10)$ for the 1-D transport example, showing that their theoretical bounds are rarely achieved in practice. This issue clearly requires further investigation and is well-beyond the scope of our paper. We hope that we have clarified what we meant in this context in our rebuttal.
> > > > > >
> > > > > > (Contd .......................................................................................................................)

---

> > > > > > > ### Author Response · Authors · 2023-08-18
> > > > > > > **Reply to the Reviewer's comments: Part IV**
> > > > > > >
> > > > > > > **Reply to Qn (6)**: We thank the reviewer for the positive comments on the numerical experiments and additions to it as outlined in the uploaded 1-page pdf. Regarding your questions about *generalization error and real world datasets*, it is a very fair point. However, such a comparison is beyond the scope of our paper as we do not propose any new architectures and only analyze existing ones. There are examples in the literature of problems where SNO, PCAnet, DeepONet either outperform or are competitive with FNO or CNN (which are not ReNOs) although deriving general conclusions about overall error are difficult as it has many components.
> > > > > > >
> > > > > > > **Reply to Qn (7)**: We thank the reviewer for your suggestion about the instability of PCA algorithm for random data. This certainly warranties further investigation although we suspect that potential non-uniqueness of empirical PCA might be implicated in these instabilities. Nevertheless, we will examine this issue carefully in a CRV, if accepted.
> > > > > > >
> > > > > > > **Reply to Qn (9)**: We apologize to the reviewer for not answering the question about convolution layer of CNO in the original rebuttal. It was simply due to the lack of space in the original rebuttal. In our understanding, the authors of the CNO paper actually prove that their convolution layer is a ReNO. The argument is provided in Appendix A.5 of the paper. We are simply copying their argument here: their *continuous convolutional operator* takes the form $\mathcal{K}_{w}f(x) = \sum k(m,n)f(x-z(m,n))$ , for all $x \in \mathbb{R}$, $k(m,n) \in \mathbb{C}$ and $z(m,n) = (m/2w,n/2w)$.
> > > > > > >
> > > > > > > Using properties of bandlimited functions, $\mathcal{K}_{w}$ is a well-defined operator from $\mathcal{B}_w(\mathbb{R}^2)$, the space of two variables bandlimited functions with bandwidth $w$, into itself. Moreover, its discretized version is defined via the last Eqn on page 20 of Raonic et al. and consequently the convolutional layer satisfies the ReNO property. The corresponding commutative diagram is given at the beginning of Page 21 of Raonic et al.
> > > > > > >
> > > > > > > On the other hand, as pointed out in Section~4 of our paper, the discrete convolution in CNNs can not be associated to any continuous operator from $\mathcal{B}_w(\mathbb{R}^2)$ into itself and the convolutional operation does not satisfy the ReNO property for any choice of the bandwidth $w$. Thus, there is a difference between the two convolutional layers.
> > > > > > >
> > > > > > > We sincerely hope that we have addressed the reviewer's detailed questions to their satisfaction in our detailed reply. If so, we kindly request the reviewer to upgrade their assessment accordingly.

---

> > > > > > > > ### Comment · Reviewer_nALh · 2023-08-20
> > > > > > > >
> > > > > > > > I thank the authors for their effort and detailed response. While I would like to continue this discussion, I don't want to hold up the publication of this paper as I do think it is a valuable contribution. I've raised my score to a 6; good luck!

---

> > > > > > > > > ### Author Response · Authors · 2023-08-20
> > > > > > > > > **Thanking the reviewer**
> > > > > > > > >
> > > > > > > > > We would like to sincerely thank the reviewer for raising our score as well as for your very valuable feedback which has enabled us to
> > > > > > > > > critically examine several aspects of our paper and will help us to strengthen the results, the message as well as the utility of this
> > > > > > > > > paper to the operator learning community. We will endeavor to incorporate these changes in a CRV, if accepted.

---

### Author Rebuttal · Authors · 2023-08-08

At the outset, we would like to thank all five reviewers for their thorough and patient reading of our article. Their criticism and constructive suggestions will enable us to improve the quality of our article. If our paper is accepted, we will incorporate all the changes that we outline below in a camera-ready version ( CRV) of our article. As many of the reviewers had questions about the experimental analysis (Sec 5), we provide a very detailed clarification below in this response. Individual comments of each reviewer is answered in the following. We have also added a 1-page pdf to supplement our argument.

Yours sincerely,

the Authors

### Clarification of Experimental Analysis Section

We wish to learn an unknown target operator $Q$ using neural operators. In this experiment, all neural operators (CNN, FNO and SNO) take as input pointwise evaluations on a grid, and are able evaluate inputs at varying grid resolutions. The goal here is to check whether their outputs at the discrete level are consistent when varying grid resolution, and verify our theory, which--informally-- tells us the outputs are consistent if and only if the operator is a ReNO.

**Construction of the target operator:** $Q: H \to H$ with $H$ being the space of periodic and $K=30$-bandlimited functions on $[-1,1]$. A simple way of constructing such a mapping is by sampling input and output pairs in a random fashion. How to sample a function? As we know that $\Psi_K := \\{ d_K(. - x_k)\\}\_{k=-K, ..., K}$ constitutes a frame for $H$, $d_K$ being the Dirichlet kernel of order $K$ and $x_k=\frac{k}{2K+1}$,
 any function $f \in H$ can be written as $f(x) = \sum_{k=-K}^{K} f(x_k) d_K(x-x_k)$. Thus, the discrete representation of $f$ simply corresponds to its point-wise evaluations on a grid, i.e. $\\{f(x_k)\\}_{k=-K, \dots, K}$. Note that for simplicity we have used and will use the same frame sequences for both the  input and output space $H$. After sampling a finite number of such input-output function pairs, we wish to learn the mapping between them using our neural operators.

**Training and Evaluation:** During training, we simply learn the neural operator $u_K: \mathbb{R}^{2K+1} \to \mathbb{R}^{2K+1}$ between frame coefficients associated to $\Psi_K$, which are the point-wise evaluations of the input-target functions that we had sampled in the previous step.  In other words, we regress to the frame coefficients of the target function, with coefficients of the input function as input to the neural operator. Once training is over, we evaluate how the different neural operators behave when dealing with changing input and output frame sequences. The frame sequences here are $\Psi_M$ for different testing frames, with associated $2M+1$ sized grids, with associated discrete operator $u_M: \mathbb{R}^{2M+1} \to \mathbb{R}^{2M+1}$.

On the continuous level, SNO maps from $H$ into $H$. Thus, at the discrete level, $u_M$ corresponds to the following: it takes in $2M+1$ point values, synthesizes these to a function in $H$ which is then sampled on the training grid. Then, $u_K$ is applied to this input vector of length $2K+1$. Finally, the output vector is synthesized to a function in $H$ and then sampled on the evaluation grid of $2M+1$ points. On the other hand, evaluation of both CNN and FNO on different grids is straightforward.

Having defined training and evaluation frame sequences, the resulting aliasing errors (See Eqn (5.1)) can now be computed with the discrete aliasing map $$\epsilon(u_K, u_M): \mathbb{R}^{2K+1} \to \mathbb{R}^{2K+1},  \epsilon(u_K, u_M) = u_K - T_{\Phi_K}^\dagger \circ T_{\Phi_M} \circ u_M \circ T_{\Psi_M}^\dagger \circ T_{\Psi_K}$$

Intuitively, $T_{\Psi_M}^\dagger \circ T_{\Psi_K}: \mathbb{R}^{2K+1} \to \mathbb{R}^{2M+1}$ first transforms input data from the training $2K+1$ grid to the evaluation $2M+1$ grid, then the output is computed using the neural operator at resolution $M$, and finally $T_{\Phi_K}^\dagger \circ T_{\Phi_M}: \mathbb{R}^{2M+1} \to \mathbb{R}^{2K+1}$ transforms outputs back from evaluation grid to training grid, so that even though $M \neq K$, discrete outputs are of a different size, these can be readily compared. This is precisely the procedure that led to the aliasing error plot of Fig. 4 of the main paper.

**Additional Experimental Follow-up**: Following the suggestions of some reviewers about adding experiments as well questions about whether FNO can be made into a ReNO, we describe a simple method to do so. We repeat the same experiment as above with FNO but we add a term to the loss function that corresponds to the discrete aliasing error $\epsilon(u_K, u_M)$, defined above. Thus, we minimize both the regression error and the aliasing error simultaneously, with a parameter $\lambda$ weighing the aliasing error. To evaluate $\epsilon(u_K, u_M)$ during training, we randomly sample $M \in [K,2K]$, evaluate the differentiable aliasing error as above and backpropagate. The results are shown in Fig. 2 of the attached pdf. They demonstrate that as $\lambda$ increases, the aliasing error is indeed minimized (Fig. 2 (Left)) and is almost negligible for $\lambda =1$, not just in the range $M \in [K,2K]$, used to evaluate aliasing error during training, but it also generalizes very well to $M > 2K$, not seen during training. Moreover, the training error (Fig. 2 (Right)) continues to be very small even when $\lambda$ is increased, showing that there is no trade-off between approximation error and aliasing error, at least for this experiment and both can be made small simultaneously for FNO.

**Are any other neural operators ReNOs ?**: Following suggestions from reviewers, we investigate PCAnet of Bhattacharya et al and were able to show it is a ReNO with the commutative diagram (Fig. 3 of attached pdf) and in Fig. 4, we also verify the ReNO property empirically in the above experiment for PCAnet and the recently introduced CNO of arXiv:2302.01178.

---

### Decision · Program_Chairs · 2023-09-21

**Decision:**

Accept (poster)

**Comment:**

The paper proposes an interesting perspective and discussion on neural operators. I found the extensive discussions between the authors and the reviewers very interesting and these really add to the paper. I believe the discussion in the paper with the agreed changes will be pertinent for the community. Given this, I recommend the paper for acceptance, conditioned on the authors making the promised changes (updated title, additional details about the experimental setting, summary figure, background on frame theory, incorporating the discussion in the rebuttal including super-resolution), as these strongly improve the paper.